# FAST FEEDFORWARD 3D GAUSSIAN SPLATTING COMPRESSION

**Yihang Chen[1,2], Qianyi Wu[2],* Mengyao Li[3,2], Weiyao Lin[1],* Mehrtash Harandi[2], Jianfei Cai[2]**
[1]Shanghai Jiao Tong University, [2]Monash University, [3]Shanghai University
{yhchen.ee, wylin}@sjtu.edu.cn, sdlmy@shu.edu.cn,
{qianyi.wu, mehrtash.harandi, jianfei.cai}@monash.edu

## ABSTRACT

With 3D Gaussian Splatting (3DGS) advancing real-time and high-fidelity rendering for novel view synthesis, storage requirements pose challenges for their widespread adoption. Although various compression techniques have been proposed, previous art suffers from a common limitation: *for any existing 3DGS, per-scene optimization is needed to achieve compression, making the compression sluggish and slow.* To address this issue, we introduce **F**ast **C**ompression of 3D **G**aussian **S**platting (FCGS), an optimization-free model that can compress 3DGS representations rapidly in a single feed-forward pass, which significantly reduces compression time from minutes to seconds. To enhance compression efficiency, we propose a multi-path entropy module that assigns Gaussian attributes to different entropy constraint paths for balance between size and fidelity. We also carefully design both inter- and intra-Gaussian context models to remove redundancies among the unstructured Gaussian blobs. Overall, FCGS achieves a compression ratio of over $20\times$ while maintaining fidelity, surpassing most SOTA per-scene optimization-based methods. Code: github.com/YihangChen-ee/FCGS.

## 1 INTRODUCTION

In recent years, 3D Gaussian Splatting (3DGS) (Kerbl et al., 2023) has significantly advanced the field of novel view synthesis. By leveraging fully explicit Gaussians with color and geometry attributes, 3DGS facilitates efficient scene training and rendering through rasterization techniques (Zwicker et al., 2001). However, the vast number of Gaussians poses a considerable storage challenge, hindering its wider application. To address the storage challenges associated with 3DGS, various compression methods have been developed, as surveyed in (Bagdasarian et al., 2024). These advancements have significantly reduced the storage requirements, bringing them to an acceptable scale. By reviewing the key developments in this area, we identified two fundamental principles that underpin the compression of 3DGS: the *value-based* and *structure-based* principles.

- *Value-based* principle. This principle assesses the importance of parameters through value-based importance scores or similarities. Techniques like pruning and vector quantization are employed to reduce the parameter count by retaining only the most representative values. With this principle, earlier studies (Lee et al., 2024; Niedermayr et al., 2024; Fan et al., 2024) focused on compactly representing unorganized Gaussian primitives based on their parameter values, and discarded less significant parameters for model efficiency.

- *Structure-based* principle. More recent approaches have shifted towards exploiting the structural relationships between Gaussian primitives for improved compression (Lu et al., 2024; Bagdasarian et al., 2024). This principle emphasizes leveraging organized structures to systematically arrange unsorted Gaussian primitives, which helps eliminate redundancies by establishing structured connections. For instance, HAC (Chen et al., 2024b) uses the Instant-NGP (Müller et al., 2022) to organize Gaussian anchor features, SOG adopts a self-organizing grid (Morgenstern et al., 2023), and IGS (Wu & Tuytelaars, 2024) uses a triplane (Chan et al., 2022) structure to efficiently organize Gaussian data.

---

*Corresponding authors.

Figure 1: **Left**: Existing compression methods require optimization of the existing 3DGS, leading to the drawback of being time-consuming for training. Our proposed FCGS overcomes this issue by compressing 3DGS representations in a single feed-forward pass, significantly reducing time consumption for compression. **Right**: Compared to Lightgaussian (Fan et al., 2024), FCGS achieves improved RD performance while requiring much less execution time on the *DL3DV-GS* dataset.

Although effective, these compression techniques have a common limitation: *for any existing 3DGS, per-scene optimization is needed to achieve compression*, as shown in Figure 1. While the optimization-based compression pipeline benefits from scene-specific training to achieve superior RD performance, it slows down the compression significantly due to time-consuming finetuning.

To address this challenge, we propose a novel approach: an optimization-free compression pipeline that enables fast compression of existing 3DGS representations through a single feed-forward pass, which we call **FCGS** (**F**ast **C**ompression of 3D **G**aussian **S**platting) from now on. Agnostic to the source of the 3DGS (either it is from optimizaiton (Kerbl et al., 2023) or from feed-forward models (Charatan et al., 2024; Szymanowicz et al., 2024; Chen et al., 2024c; Tang et al., 2024)), our FCGS allows for fast compression, offering a convenient and hassle-free solution. In contrast to previous methods that degrade the parameter values or alter the 3DGS structure, thereby requiring additional finetuning, FCGS aims to preserve both the values and the structure integrity of 3DGS, enabling an optimization-free design. The Multi-path Entropy Module (MEM), designed under the *value-based* principle, deduces masks that determine whether attributes should be directly quantized for compression or processed through an autoencoder. Building on the masks determined by MEM and inspired by the *structure-based* principle, we propose both inter- and intra-Gaussian context models that effectively capture structural relationships among Gaussian attributes.

It is important to highlight that both pipelines of ***per-scene optimization-based compression*** (previous methods) and ***generalizable optimization-free compression*** (our approach) are significant and serve different purposes. The former benefits from per-scene adaptation to achieve superior RD performance but suffers from slow compression speed, making it suitable for permanent data storage or server-side encoding. In contrast, our pipeline offers a convenient and hassle-free solution, where a single model can *directly* and *rapidly* compress various 3DGS without the need for finetuning, making it well-suited for time-sensitive applications. To the best of our knowledge, our work is the first to achieve a ***generalizable optimization-free compression pipeline*** for 3DGS. Although the absence of per-scene finetuning naturally limits our RD performance, we have still achieved a compression ratio exceeding $20\times$ while maintaining excellent fidelity by meticulously designing MEM and context models. Our contributions are summarized as follows:

- We propose a pioneering fast and generalizable optimization-free compression pipeline for 3D Gaussian Splatting, named FCGS, effectively broadening the application scenarios for 3DGS compression techniques.
- To facilitate efficient compression of 3DGS representations, we introduce the MEM module to balance size and fidelity across different Gaussians. Additionally, we meticulously customize both inter- and intra-Gaussian context models, significantly enhancing compression performance by effectively eliminating redundancies.
- Extensive experiments across various datasets demonstrate the effectiveness of FCGS, achieving a compression ratio of $20\times$ while maintaining excellent fidelity, even surpassing most of the optimization-based methods.

## 2 RELATED WORK

The field of 3D scene representation for 3D scenes has seen significant advancements in recent years. Neural Radiance Field (NeRF) (Mildenhall et al., 2021) models scene information using fully

implicit MLPs, which predict opacity and RGB values at 3D coordinates for ray rendering. However, these MLPs are designed to be quite large to capture all the scene-specific details, leading to slow training and rendering times. To improve both fidelity and rendering efficiency, subsequent works (Müller et al., 2022; Chen et al., 2022; Sun et al., 2022) have introduced learnable explicit representations that assign scene-specific information to the input coordinates before they are processed by the MLPs. These methods, however, increase storage requirements due to the use of explicit representations. 3D Gaussian Splatting (Kerbl et al., 2023) takes this further by representing the scene entirely through explicit attributed Gaussians. Using rasterization techniques, these Gaussians can be quickly rendered into 2D images for a given viewpoint. Unfortunately, this fully explicit representation significantly increases storage demands. To address the storage issues of NeRF and 3DGS, various research efforts have focused on compression techniques, which generally fall into two main design principles: *value-based* and *structure-based*.

The *value-based* principle focuses on the importance scores or similarities of parameters. Using these scores or similarities, pruning and vector quantization can be applied via thresholding or clustering, allowing for compression by retaining only the most representative parameters. However, this process often alters the original parameter values by a noticeable margin, necessitating an additional finetuning step to recover the lost information. For NeRF, methods like VQRF (Li et al., 2023) and Re:NeRF (Deng & Tartaglione, 2023) calculate the significance of parameters based on opacity or gradient values and apply pruning or vector quantization, followed by a finetuning phase. BiRF Shin & Park (2023) investigates parameters' values and binarizes them. For 3DGS, similar value-based thresholding and clustering techniques are commonly used, as seen in works such as Lee et al. (2024); Niedermayr et al. (2024); Navaneet et al. (2024); Fan et al. (2024); Girish et al. (2024); Wang et al. (2024a); Ali et al. (2024); Fang & Wang (2024), which consider Gaussian values. Additionally, scalar quantization and knowledge distillation are implemented in Girish et al. (2024) and (Fan et al., 2024), respectively.

The *structure-based* principle represents another major direction that investigates the structural redundancies of parameters. CNC (Chen et al., 2024a) introduced a pioneering proof of concept by applying level- and dimension-wise context models to compress hash grids for Instant-NGP (Müller et al., 2022) in the NeRF series. This structure-based design is also evident in works like Tang et al. (2022); Rho et al. (2023); Girish et al. (2023); Li et al. (2024). CodecNeRF (Kang et al., 2024) investigates a feed-forward approach to generate neural radiance field for scenes and then directly compresses them. In the case of 3DGS, structure-based compression methods have also seen notable progress. The primary challenge is the sparse and unorganized nature of 3DGS, which makes it difficult to identify and utilize structural relationships. Grid-based sorting (Morgenstern et al., 2023) addresses this by projecting 3D Gaussians onto 2D planes for compression while preserving spatial relationships. Scaffold-GS (Lu et al., 2024) uses anchors to cluster nearby Gaussians that share common features, and based on this, works such as Chen et al. (2024b); Liu et al. (2024); Wang et al. (2024b) further reduce redundancies for anchors, resulting in improved compression performance. HAC (Chen et al., 2024b) and IGS (Wu & Tuytelaars, 2024) explore relations among the organized grids and Gaussians. SUNDAE (Yang et al., 2024) employs spectral Graph to improve pruning.

However, these approaches all require per-scene finetuning for compression on a given 3DGS, which can be time-consuming. In this paper, we explore a novel, optimization-free pipeline for 3DGS compression, innovatively building context models for Gaussian primitives, which are essential for eliminating redundancy and improving compression efficiency.

## 3 FAST COMPRESSION OF 3D GAUSSIAN SPLATTING

Our goal is to rapidly compress a 3DGS representation in a single feed-forward pass without finetuning. Inspired by image compression methods (Ballé et al., 2018), we adopt an autoencoder-based structure, where the Gaussian attributes are encoded into a latent space for compression, as shown in Figure 2. However, we found that treating all Gaussian parameters equally and feeding them all into the same autoencoder could lead to a significant loss of fidelity, as some attributes were highly sensitive to deviations. To address this, we introduce a Multi-path Entropy Module (MEM) to effectively balance compression size and fidelity. For the context model design, we customize both inter- and intra-Gaussian context models that effectively eliminate redundancies among parameters of Gaussians (Figure 3), which are lightweight and efficient. Additionally, we use a Gaussian

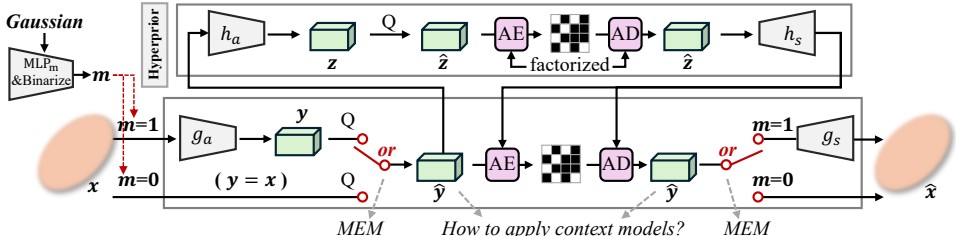

Figure 2: Our approach is inspired by image compression, where the input Gaussian attributes $x$ is mapped into the latent space $\hat{y}$ for compression after passing through an analysis transform $g_a$ and quantization to eliminate redundancies. To compress $\hat{y}$, a hyperprior branch is introduced, using the coarse representation $\hat{z}$ to estimate the distribution parameters of $\hat{y}$ under a Gaussian distribution assumption, which aids in entropy encoding and decoding. In addition to the hyperprior, various context models are applied to $\hat{y}$ to improve the estimation of distribution probabilities. After decoding $\hat{y}$, a synthesis transform $g_s$ projects it back to the original space as $\hat{x}$. A loss function is used to maintain high fidelity between $\hat{x}$ and $x$ using their rendered images, while minimizing the entropy of $\hat{y}$ and $\hat{z}$. AE and AD represent Arithmetic Encoding and Arithmetic Decoding, respectively. In our paper, we implement transform networks as simple MLPs. $g_a$ and $g_s$ consist of 4 layers each, while $h_a$ and $h_s$ have 3 layers each.

Mixture Model (GMM) to estimate the probability distribution. In the following sections, we first discuss the background and characteristics of 3DGS, and then delve into the design of our FCGS.

## 3.1 Preliminaries and Discussions

**3D Gaussian Splatting (3DGS)** (Kerbl et al., 2023) employs a large number (*e.g.*, 1 million) attributed Gaussians to represent a 3D scene. These Gaussians are characterized by color and geometry parameters, and are splatted along given views using rasterization to generate rendered images. Each Gaussian is defined by a covariance $\boldsymbol{\Sigma} \in \mathbb{R}^{3\times3}$ and a location (mean) $\boldsymbol{\mu}^{\mathrm{g}} \in \mathbb{R}^3$,

$$G(\boldsymbol{l}) = \exp\left(-\frac{1}{2}(\boldsymbol{l} - \boldsymbol{\mu}^{\mathrm{g}})^\top \boldsymbol{\Sigma}^{-1}(\boldsymbol{l} - \boldsymbol{\mu}^{\mathrm{g}})\right), \tag{1}$$

where $\boldsymbol{l} \in \mathbb{R}^3$ is a random 3D location, and $\boldsymbol{\Sigma}$ is defined by a scaling matrix $\boldsymbol{S} \in \mathbb{R}^{3\times3}$ and a rotation matrix $\boldsymbol{R} \in \mathbb{R}^{3\times3}$ such that $\boldsymbol{\Sigma} = \boldsymbol{R}\boldsymbol{S}\boldsymbol{S}^\top\boldsymbol{R}^\top$. To render a pixel value $\boldsymbol{C} \in \mathbb{R}^3$, the Gaussians are first splatted to 2D, and rendering is performed as follows:

$$\boldsymbol{C} = \sum_i \boldsymbol{c}_i \alpha_i \prod_{j=1}^{i-1}(1 - \alpha_j) \tag{2}$$

where $\alpha \in \mathbb{R}$ is the opacity, and $\boldsymbol{c} \in \mathbb{R}^3$ is the view-dependent color of each Gaussian, which is calculated from 3-degree Spherical Harmonics (SH) $\in \mathbb{R}^{48}$. In this paper, we design our FCGS model based on this SH-based rendering structure of 3DGS.

**Discussions.** The attributes of Gaussians can be categorized into *geometry* and *color* attributes. The *geometry* attributes (opacity $\boldsymbol{\alpha}$, diagonal elements of scaling $\boldsymbol{S}$, and quaternion representation of rotation $\boldsymbol{R}$) determine the dependencies of the rasterization process (*e.g.*, the coverage range of the Gaussians or the depth to which they should be composed). Once the geometric dependencies are established, the *color* attributes are used to assign colors via 3-degree SH following (Kerbl et al., 2023). Since the *geometry* attributes directly influence the rasterization dependencies, they are more sensitive to deviations than the *color* attributes. For brevity, we denote *geometry* attributes as $\boldsymbol{f}^{\mathrm{geo}} \in \mathbb{R}^8$, *color* attributes as $\boldsymbol{f}^{\mathrm{col}} \in \mathbb{R}^{48}$, and the concat of both as $\boldsymbol{f}^{\mathrm{gau}} \in \mathbb{R}^{56}$. In the following sections, unless otherwise specified, $x$ refers to either $\boldsymbol{f}^{\mathrm{geo}}$ or $\boldsymbol{f}^{\mathrm{col}}$.

## 3.2 Value-Based Principle: Multi-path Entropy Module (MEM)

As shown in Figure 2, we draw inspiration from image compression and apply an autoencoder-based compression approach with a hyperprior structure, where the Gaussian attributes $x$ are projected into a quantized latent space $\hat{y}$ for compression. After obtaining $\hat{y}$ through the arithmetic decoding, a synthesis transform $g_s$ is employed to project it back to the original space $\hat{x}$, which is the decoded

version. However, in 3DGS representations, Gaussians are not the final output; the actual results are the rendered images obtained through rasterization. Unfortunately, deviations in Gaussian attributes can be amplified during the rasterization process, leading to even greater deviations in the rendered images. Some Gaussians are highly sensitive to these deviations, as they may occupy crucial positions that significantly affect the rasterization process. Simply feeding all attributes into the autoencoder does not yield satisfactory results: The forward pass of MLPs is inherently non-invertible due to non-linear activation functions, which means projecting $x$ into latent space via an MLP does not allow for exact recovery of the original $x$ from the latent. As a result, MLP-decoded attributes $\hat{x}$ cannot always align precisely with the original attributes for each Gaussian. This misalignment would be amplified in rendering, leading to a significant fidelity drop in the rendered images, as exhibited in the ablation study (Subsection 4.3).

Geometry attributes are the most sensitive to such deviations because they directly impact dependencies during rasterization. For example, if the decoded opacity of a Gaussian is too large, it could block Gaussians behind it, preventing them from contributing as they should. Similarly, deviations in scaling and rotation can have significant effects. To address this, we remove $g_a$ and $g_s$ for all geometry attributes $f^{\text{geo}}$, ensuring that $y = x$. Fortunately, color attributes are less sensitive to deviations, although they still impact the final result. For $f^{\text{col}}$, as depicted in Figure 2, we introduce MEM to deduce a binary mask $m \in \mathbb{R}$ that adaptively determines whether an $x$ (only for $f^{\text{col}}$ here) should pass through the MLPs $g_a$ and $g_s$ to eliminate redundancies, resulting in $y = g_a(x)$, or simply bypass the MLPs, setting $y = x$ to best preserve the original information,

$$\hat{x}_i = g_s(\mathbb{Q}(g_a(x_i))) \times m_i + \mathbb{Q}(x_i) \times (1 - m_i), \ m_i = \text{Sig}(\text{MLP}_{\text{m}}(f_i^{\text{gau}})) > \epsilon_m \quad (3)$$

where $\mathbb{Q}$ represents the quantization operation with a quantization step of $q$: if $y_i$ is in the latent space ($m_i = 1$), $q$ is 1; otherwise ($m_i = 0$), $q$ becomes a trainable decimal parameter. During training, uniform noise is added, while during testing, quantization is applied. $\text{Sig}$ is the Sigmoid function and $\epsilon_m$ is a hyperparameter that defines the threshold for binarizing the mask. Gradients of the binarized mask $m$ are backpropagated using STE (Bengio et al., 2013; Lee et al., 2024).

Balancing the mask rate is crucial, as it directly impacts the RD trade-off. Previous works like Lee et al. (2024) incorporate an additional loss term with a hyperparameter $\lambda_m$ to regulate the mask rate. However, given that the RD loss already includes a trade-off hyperparameter $\lambda$ to balance bits and fidelity, introducing another parameter, $\lambda_m$, would unnecessarily complicate the process of finding the optimal RD trade-off. To resolve this, we integrate the mask information directly into the bit consumption calculation, allowing the model to adaptively learn the optimal mask rate, thereby eliminating $\lambda_m$. Please refer to Subsection 3.4 for details.

### 3.3 STRUCTURE-BASED PRINCIPLE: AUTOREGRESSIVE CONTEXT MODELS

To further enhance the accuracy of probability estimation, context models play a crucial role for $\hat{y}$. Specifically, a portion of $\hat{y}$ is first decoded and then used to assist in predicting the remaining part based on contextual relationships. In image compression, where data like $\hat{y}$ are arranged in a structured grid (*i.e.*, a feature map), designing such context models is relatively straightforward. However, Gaussians in 3DGS are sparse and unorganized (as is $\hat{y}$), making it a challenge to design efficient and lightweight context models. To tackle this, we analyze the unique properties of Gaussians and develop customized inter- and intra-Gaussian context models, as illustrated in Figure 3.

**Inter-Gaussian Context Models.** 3D Gaussians collectively represent the scene, leading to inherent relationships among them. To uncover these relationships, we refer to grid-based structures, which are typically compact, organized, and capable of constructing spatial connections among Gaussians. While this "grid" structure does not naturally exist within 3DGS, we propose an innovative method to *create grids* from Gaussians themselves. Note that a concurrent work Wang et al. (2024b) proposes structuring anchors into multiple levels to model entropy. However, its anchor location quantization process introduces deviations, which impacts fidelity and requires finetuning. Differently, our inter-Gaussian context model overcomes this limitation by creating grids in a feed-forward manner for context modeling, without modifying Gaussian locations, eliminating the need of finetuning.

We divide all Gaussians' $\hat{y}$ into $N^{\text{s}}$ batches ($N^{\text{s}} = 2$ as an example in Figure 3 mid) and decode each batch sequentially. During this process, the previously decoded $\hat{y} \in \mathbb{Y}_{[0,n^{\text{s}}-1]}$ are first used to create grids, which then provide context for the to-be-decoded $\hat{y} \in \mathbb{Y}_{[n^{\text{s}}]}$ via interpolation. Here, $\mathbb{Y}_{[n^{\text{s}}]}$ refers to the set of $\hat{y}$ belonging to the $n^{\text{s}}$-th batch out of $N^{\text{s}}$. To generate the feature $f^v$ for a

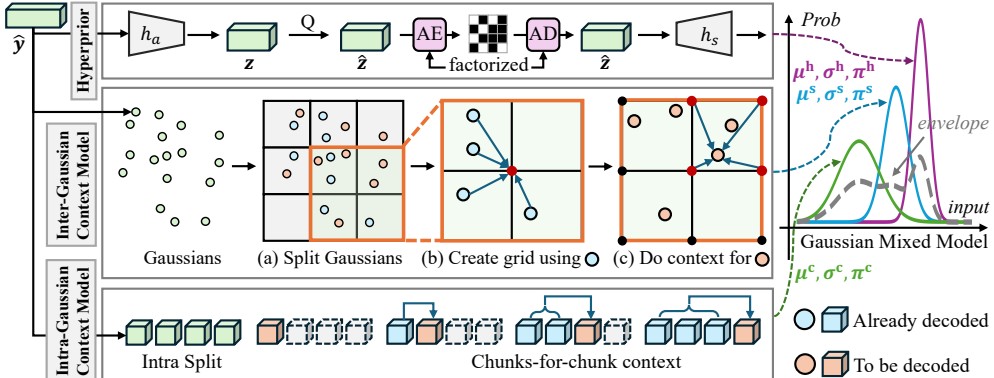

Figure 3: Context models of FCGS. We build on the hyperprior design of image compression (**top**) and introduce our inter- and intra-Gaussian context models (**mid & bottom**). Together, they form a GMM that provides a more accurate estimation of the value distribution probability of $\hat{y}$ (**right**).

voxel at position $v$ on the grids, we investigate $\hat{y}$ within $v$'s interpolation range. For a voxel located at $v_i \in \mathbb{R}^3$ (*i.e.*, the red point in Figure 3 mid (b)), its feature $f_i^v$ is computed as follows:

$$f_i^v = \frac{\sum_{k:\hat{y}_k \in \mathbb{Y}_{[0,n^s-1]}^{v_i}} w_k \hat{y}_k}{\sum_{k:\hat{y}_k \in \mathbb{Y}_{[0,n^s-1]}^{v_i}} w_k}, \quad \text{where } w_k = \prod_{\dim \in \{x,y,z\}} \left(1 - |\boldsymbol{\mu}_{k,\dim}^g - v_{i,\dim}|\right) \quad (4)$$

where $\mathbb{Y}_{[0,n^s-1]}^{v_i}$ denotes the subset of $\mathbb{Y}_{[0,n^s-1]}$ where $\hat{y}$ fall into $v_i$'s interpolation range (*i.e.*, the orange box in Figure 3 mid (a,b,c)). $k$ is the index of $\hat{y}_k$, and $w_k$ represents the weight that determines the contribution of each $\hat{y}_k$. The closer $\hat{y}_k$ is to $v_i$, the greater its contribution. Regardless of the number of $\hat{y}$ within $v_i$'s interpolation range, we can always compute a weighted average feature $f_i^v$ for $v_i$ that integrates information for interpolation in the next step.

By conducting this calculation for all the voxels, we can create the grids, which are organized and structured, enabling them to provide contextual information for any input coordinates via interpolation. For a to-be-decoded $\hat{y}_i \in \mathbb{Y}_{[n^s]}$, located at $\boldsymbol{\mu}_i^g$, its distribution parameters $\boldsymbol{\mu}_i^s$, $\boldsymbol{\sigma}_i^s$, and $\boldsymbol{\pi}_i^s$ can be computed from these grids using $\text{MLP}_s$:

$$\boldsymbol{\mu}_i^s, \boldsymbol{\sigma}_i^s, \boldsymbol{\pi}_i^s = \text{MLP}_s(\oplus[f_i^{\boldsymbol{\mu}^g}, \text{emb}(\boldsymbol{\mu}_i^g)]), \quad \text{where } f_i^{\boldsymbol{\mu}^g} = \sum_{k:v_k \in \mathbb{V}^{\boldsymbol{\mu}_i^g}} w_k f_k^v \quad (5)$$

where $\mathbb{V}^{\boldsymbol{\mu}_i^g}$ represents the set of voxels forming $\boldsymbol{\mu}_i^g$'s minimum bounding box (*i.e.*, the 4 red points in Figure 3 mid (c)), and $f_i^{\boldsymbol{\mu}^g}$ is the feature obtained by grid interpolation. $\text{emb}$ and $\oplus$ are sin-cos positional embedding and channel-wise concatenate operation, respectively. The calculated parameters $\boldsymbol{\mu}_i^s$, $\boldsymbol{\sigma}_i^s$, and $\boldsymbol{\pi}_i^s$ are then used to estimate the distribution of $\hat{y}_i$ in GMM.

In practice, we use both 3D and 2D grids, designed with multiple resolutions to capture varying levels of detail. The 3D grids effectively represent the spatial relationships between the Gaussians and the voxels, though they are of lower resolution due to the higher dimensionality. To complement this, we employ three 2D grids, each created by collapsing one dimension. These 2D grids provide higher resolutions, allowing for a more precise capture of local details.

**Intra-Gaussian Context Models.** Beyond addressing redundancies across Gaussians, we further utilize our FCGS to eliminate redundancies within each Gaussian. Specifically, we split each $\hat{y}_i$ into $N^c$ chunks, and $\text{MLP}_c$ is used to deduce the distribution parameters for each chunk from the previously decoded ones. For path $m = 1$ in MEM, we split $\hat{y}$ into $N^c = 4$ chunks along channel dimension. For path $m = 0$, $\hat{y}$ is split into $N^c = 3$ chunks along RGB axis to leverage redundancies of color components. We do not apply intra-context for $f^{\text{geo}}$, as its internal relations are trivial.

$$\boldsymbol{\mu}_i^c, \boldsymbol{\sigma}_i^c, \boldsymbol{\pi}_i^c = \oplus_{n^c=1}^{N^c} \{\boldsymbol{\mu}_{i,n^c}^c, \boldsymbol{\sigma}_{i,n^c}^c, \boldsymbol{\pi}_{i,n^c}^c\}, \quad \text{where } \boldsymbol{\mu}_{i,n^c}^c, \boldsymbol{\sigma}_{i,n^c}^c, \boldsymbol{\pi}_{i,n^c}^c = \text{MLP}_c(\hat{y}_{i,[0,n^c c - c)}) \quad (6)$$

where $n^c$ and $c$ are chunk index and channel amount per chunk, respectively. $\boldsymbol{\mu}_{i,n^c}^c, \boldsymbol{\sigma}_{i,n^c}^c, \boldsymbol{\pi}_{i,n^c}^c$ are the intermediate probability parameters for chunk $n^c$, with a dimension size of $c$ for each.

**Gaussian Mixed Model (GMM).** In addition to the context models, hyperprior also outputs a set of distribution parameters $\boldsymbol{\mu}_i^h$, $\boldsymbol{\sigma}_i^h$ and $\boldsymbol{\pi}_i^h$ from $\hat{z}_i$. To this end, we introduce GMM to represent the

final probability as combination of these 3 sets of Gaussian distribution parameters. For any $\hat{\boldsymbol{y}}_i$ at channel $j$, its probability can be calculated as combination of cumulative Gaussian distribution $\Phi$,

$$p(\hat{y}_{i,j}) = \sum_{l \in \{\text{h,s,c}\}} \theta^l_{i,j} \left( \Phi(\hat{y}_{i,j} + \frac{q^l}{2} \mid \mu^l_{i,j}, \sigma^l_{i,j}) - \Phi(\hat{y}_{i,j} - \frac{q^l}{2} \mid \mu^l_{i,j}, \sigma^l_{i,j}) \right), \theta^l_{i,j} = \frac{\exp(\pi^l_{i,j})}{\sum_{v \in \{\text{h,s,c}\}} \exp(\pi^v_{i,j})}$$

(7)

Note that for **geometry** attributes $\boldsymbol{f}^{\text{geo}}$, $l \in \{\text{h,s}\}$. GMM adaptively weights the mixing of the three sets of distributions, providing more accurate probability estimation. For $\hat{\boldsymbol{z}}$ in the hyperprior branch, we follow Ballé et al. (2018) to use a factorized module to estimate its $p(\hat{z}_{i,j})$. According to information theory (Cover, 1999), bit consumption of $\boldsymbol{x}_i$ can be calculated: $bit_i = \sum_{j}^{D^y} (-\log_2 p(\hat{y}_{i,j})) + \sum_{j}^{D^z} (-\log_2 p(\hat{z}_{i,j}))$. $D^y$ and $D^z$ are number of channels of $\hat{\boldsymbol{y}}_i$ and $\hat{\boldsymbol{z}}_i$.

## 3.4 TRAINING AND CODING PROCESS

**Training loss.** After obtaining $\hat{\boldsymbol{x}}$ from $\hat{\boldsymbol{y}}$ through the synthesis transform $g_s$, we can evaluate the fidelity between the ground truth $\boldsymbol{I}$ and the image $\hat{\boldsymbol{I}}$ rendered from $\hat{\boldsymbol{x}}$. To balance the mask rate of $\boldsymbol{f}^{\text{col}}$ in MEM, we include the mask $m$ directly in the bit calculation process, rather than introducing an additional loss term. Specifically, we forward *all* Gaussians' $\boldsymbol{f}^{\text{col}}$ in both MEM paths and use $m$ to compute the weighted sum of the output bit from both paths, ensuring the gradients can be backpropagated for both. The overall loss function is:

$$L = L_{\text{fidelity}}(\hat{\boldsymbol{I}}, \boldsymbol{I}) + \lambda \frac{L_{\text{entropy}}}{N^g \times 56}, \text{ where } L_{\text{entropy}} = \sum_{i=1}^{N^g} bit_i^{\text{geo}} + m_i bit_i^{\text{col}|m_i=1} + (1 - m_i) bit_i^{\text{col}|m_i=0}$$

(8)

where $N^g$ represents the amount of Gaussians, 56 is the dimension of $\boldsymbol{f}^{\text{gau}}$, and they collectively represent the amount of attribute parameters. $bit_i^{\text{geo}}$ and $bit_i^{\text{col}}$ represents the bit consumption for a Gaussian's **geometry** and **color** attributes, respectively, with $\boldsymbol{x}_i$ set as $\boldsymbol{f}_i^{\text{geo}}$ or $\boldsymbol{f}_i^{\text{col}}$, respectively. $\lambda$ is a hyperparameter controlling the trade-off between fidelity and entropy. The fidelity loss, $L_{\text{fidelity}}$ includes both MSE and SSIM metrics to evaluate the reconstruction quality of $\hat{\boldsymbol{x}}$. Minimizing this loss encourages $\hat{\boldsymbol{x}}$ to closely resemble the original $\boldsymbol{x}$ while also reducing the bit consumption.

**Encoding/decoding process. For encoding**, the attributes $\boldsymbol{f}^{\text{geo}}$ and $\boldsymbol{f}^{\text{col}}$ are encoded using Arithmetic Encoding (AE) (Witten et al., 1987) in the space of $\hat{\boldsymbol{y}}$ and $\hat{\boldsymbol{z}}$ given the corresponding probabilities, masks $m$ are binary and encoded using AE based on the occurrence frequency of 1, and Gaussian coordinates $\boldsymbol{\mu}^g$ are 16-bit quantized and encoded losslessly using GPCC (Chen et al., 2023). **For decoding**, Gaussian coordinates $\boldsymbol{\mu}^g$, masks $m$, and hyperpriors $\hat{\boldsymbol{z}}$ are decoded first. Then, to decode $\hat{\boldsymbol{y}}_{i,[n^c c - c, n^c c)} \mid \hat{\boldsymbol{y}}_i \in \mathbb{Y}_{[n^s]}$, its value distribution is calculated using GMM based on: $\boldsymbol{\mu}^s_{i,n^c}, \boldsymbol{\sigma}^s_{i,n^c}, \boldsymbol{\pi}^s_{i,n^c}$ from $\hat{\boldsymbol{y}} \in \mathbb{Y}_{[0,n^s-1]}$ using inter-Gaussian context models (channel sliced), and $\boldsymbol{\mu}^c_{i,n^c}, \boldsymbol{\sigma}^c_{i,n^c}, \boldsymbol{\pi}^c_{i,n^c}$ from $\hat{\boldsymbol{y}}_{i,[0,n^c c - c)} \mid \hat{\boldsymbol{y}}_i \in \mathbb{Y}_{[n^s]}$ using intra-Gaussian context models, and $\boldsymbol{\mu}^h_{i,n^c}, \boldsymbol{\sigma}^h_{i,n^c}, \boldsymbol{\pi}^h_{i,n^c}$ from $\hat{\boldsymbol{z}}_i$ using hyperprior (channel sliced). On obtaining GMM, it is decoded using AD.

# 4 EXPERIMENTS

## 4.1 IMPLEMENTATION DETAILS

**Implementation.** Our FCGS model is implemented using the PyTorch framework (Paszke et al., 2019) and trained on a single NVIDIA L40s GPU. The dimension of $\hat{\boldsymbol{y}}$ is set to 256 for **color** ($m = 1$). For $\hat{\boldsymbol{z}}$, dimensions are set to 16, 24, and 64 for **geometry**, **color** ($m = 0$), and **color** ($m = 1$), respectively. Grid resolutions are $\{70, 80, 90\}$ for 3D grids and $\{300, 400, 500\}$ for 2D grids. We set $N^s$ to 4, using uneven splitting ratios of $\{\frac{1}{6}, \frac{1}{6}, \frac{1}{3}, \frac{1}{3}\}$, with uniform random sampling. $\epsilon_m$ is set to 0.01. In inference, we maintain a same random seed in encoding and decoding to guarantee consistency. The training batch size is 1 (*i.e.*, one 3DGS scene per training step). We adjust $\lambda$ from $1e - 4$ to $16e - 4$ to achieve variable bitrates. During training, we first train $g_a$ and $g_s$ of $m = 1$ using only the fidelity loss to ensure satisfactory reconstruction quality. We then jointly train the context models for **color** with $m = 1$, and finally, train the entire model in an end-to-end manner. We adopt this training process because the gradient chain for the $m = 0$ path is shorter than that of the $m = 1$ path. Without sufficient pre-training of the $m = 1$ path, the model is prone to collapsing into a local minimum where all $m$ values are zero.

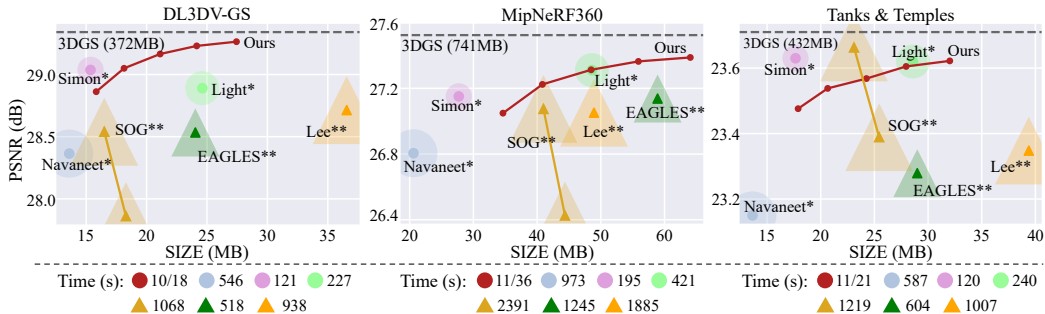

Figure 4: Performance comparison. Each scene is initially trained for $30K$ iterations to produce the vanilla 3DGS. Methods marked with * and circle are finetuned from this common 3DGS (our FCGS also compresses the same 3DGS); Methods marked with ** and triangles are trained from scratch due to their modification to structures. We also present the runtime of our method and other approaches at the **bottom** of the figure (which is also reflected by the size of the marks), where our approach requires significantly less time for compression. For our runtime, it means using multiple/single GPUs. Thanks to our optimization-free pipeline, we divide the 3DGS into chunks, with each chunk containing 1 million Gaussians, allowing us to easily encode these chunks in parallel using multiple GPUs, further speeding up the process.

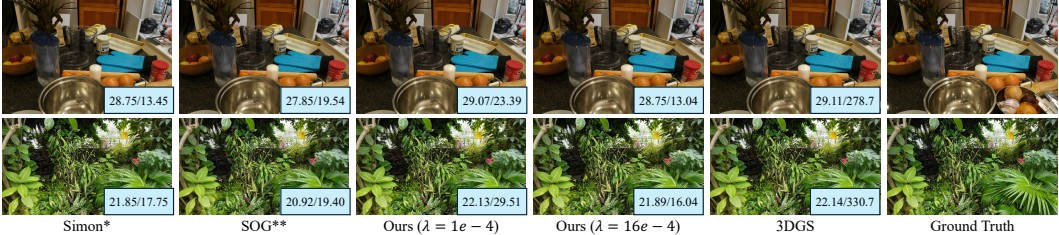

Figure 5: Qualitative comparison. We achieve substantial size reduction while preserving high fidelity. PSNR (dB) / SIZE (MB) are indicated in the bottom-right corner. We only present two baseline methods for qualitative comparison here due to space limitation. Please refer to Appendix Section N for more comprehensive qualitative comparisons.

**Training Dataset.** FCGS requires training on a large-scale dataset with abundant 3DGS. To achieve that, we refer to *DL3DV* dataset (Ling et al., 2024), which contains approximately $7K$ multi-view scenes. We train these scenes to generate their 3DGS at a resolution of 960P, which takes about 60 GPU days on NVIDIA L40s. After filtering out low-quality ones, we obtain 6770 3DGS, and randomly split 100 for testing and the remaining for training. This dataset is referred to as *DL3DV-GS*. For more details regarding *DL3DV-GS*, please refer to Appendix Section A.

**Metrics.** We assess compression performance in terms of fidelity relative to size. Herein, we present PSNR metric to evaluate fidelity due to page constraints. For additional metrics (SSIM (Wang et al., 2004) and LPIPS (Zhang et al., 2018)), please refer to Appendix Section F.

## 4.2 EXPERIMENT EVALUATION

For 3DGS from optimization, FCGS enables rapid compression. Since no prior works customize optimization-free compression for 3DGS, direct comparisons are unavailable. This leaves us to benchmark against optimization-based methods, a comparison that is inherently unfair to FCGS. Nonetheless, we achieve excellent RD performance despite this lack of optimization. Furthermore, FCGS can also compress 3DGS generated by feed-forward models (Chen et al., 2024c; Tang et al., 2024), demonstrating its versatility.

**For 3DGS from optimization**, we employ *DL3DV-GS*, *MipNeRF360* (Barron et al., 2022), and *Tank&Temples* (Knapitsch et al., 2017) for evaluation. For the baseline methods, we compare those built upon the vanilla 3DGS structure, including both finetune from existing 3DGS (Navaneet et al.,

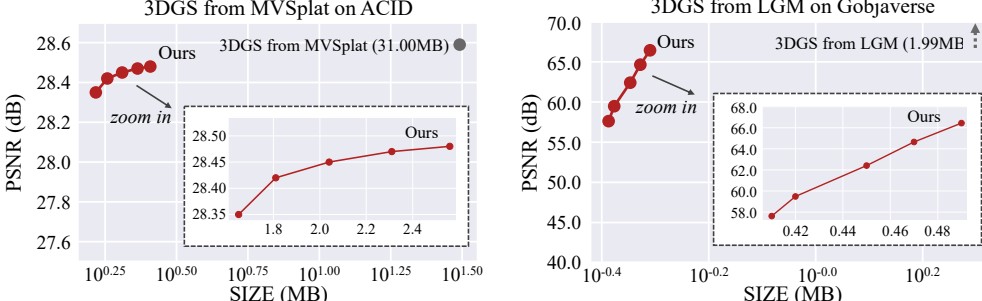

Figure 6: Compression of 3DGS from feed-forward models. **Left**: MVSplat is a generalizable reconstruction model, whose fidelity is measured between the rendered images and the ground truth. **Right**: LGM is a generative model, whose fidelity is measured between images rendered from 3DGS before and after compression, as no ground truth cannot be measured due to its generative nature.

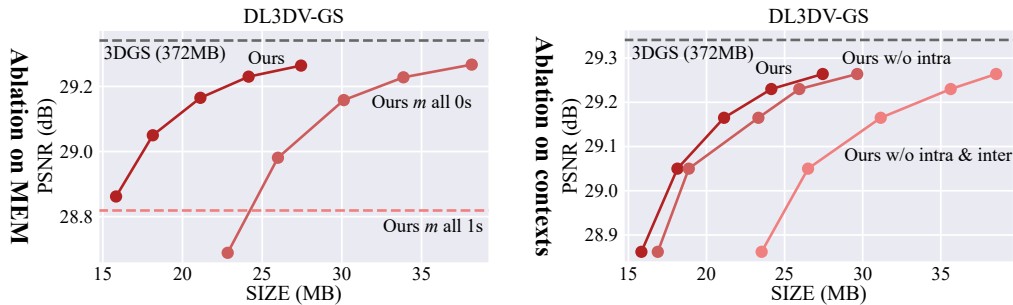

Figure 7: Ablation studies on the *DL3DV-GS* dataset. **Left**: Setting $m$ to all 0s results in a significant increase in bit consumption. Conversely, setting $m$ to all 1s leads to a drastic drop in fidelity, even without applying quantization or entropy constraints to $\boldsymbol{y}$. **Right**: Excluding the proposed context models leads to a substantial increase in bit consumption, due to the absence of mutual dependencies.

2024; Niedermayr et al., 2024; Fan et al., 2024) or train from scratch (Morgenstern et al., 2023; Girish et al., 2024; Lee et al., 2024). The results are in Figure 4. Although our FCGS lacks per-scene adaptation, which naturally puts us at a disadvantage for unfair comparison, it still surpasses most optimization-based methods, thanks to the effectiveness of MEM and context models. The qualitative comparisons are shown in Figure 5, which exhibit high fidelity after compression. Please refer to Appendix Section D and G to find discussions on the SoTA compression methods and the detailed analysis on the storage size of each component, respectively.

**For 3DGS from feed-forward models**, we also demonstrate compression capability. To evaluate our performance, we refer to MVSplat (Chen et al., 2024c) and LGM (Tang et al., 2024). MVSplat is a generalizable model that reconstructs interpolated novel views given two bounded views. LGM is a generative model that creates the 3D scenes from the 4 multi-view images. We follow LGM to generate the remaining 3 views from the initial 1 view using (Wang & Shi, 2023), which are then collectively input into LGM for 3DGS creation. This process results in ground-truth cannot be measured for LGM, thus we evaluate the compression fidelity by measuring the similarities of images rendered from 3DGS before or after compression. We utilize 10 scenes from *ACID* (Liu et al., 2021) and 50 scenes from *Gobjaverse* (Qiu et al., 2023; Deitke et al., 2022) for these two models for evaluation. The results are shown in Figure 6. Although trained on 3DGS from optimization (Kerbl et al., 2023), FCGS still generalizes in a zero-shot manner to feed-forward-based 3DGS, achieving compression ratios of $15\times$ and $5\times$. Notably, when compressing 3DGS from feed-forward models, we set mask $m$ to all 0s for *color* attributes. Please refer to Appendix Section B for limitation analysis.

### 4.3 ABLATION STUDY

We perform ablation studies to evaluate the effectiveness of our proposed **MEM** and **context models**. First, we investigate the **MEM** module, which adaptively selects high-tolerance *color* attributes

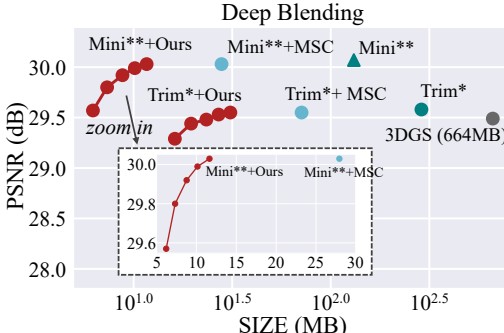

Figure 8: Building on pruning approaches, FCGS can further boost compression performance. The size of the compressed 3DGS using FCGS (at the highest rate) is only $40\%$ that of MSC with the same fidelity. When combined with pruning techniques, FCGS achieves a compression ratio of $100\times$ over the vanilla 3DGS (see Mini**+Ours). Experiments are conducted on the *Deep Blending* dataset. The x-axis uses a `Log10` scale.

to eliminate redundancies via an autoencoder structure, while ensuring all ***geometry*** attributes are directly quantized. Without MEM, we observe the following: **1)** If both color and geometry attributes are processed through the autoencoder, the model collapses because rasterization cannot be performed correctly due to deviations in the geometry information. **2)** Fixing the ***geometry*** attributes to be directly quantized, we allow all ***color*** attributes to be processed by the autoencoder (*i.e.*, all $m = 1$). As shown in Fig 7 left, even without quantization or entropy constraints on $\boldsymbol{y}$, the rendered images suffer from a significant drop in fidelity due to deviations brought by MLPs. **3)** On the other hand, if all ***color*** attributes are directly quantized (*i.e.*, all $m = 0$), their redundancies are not effectively eliminated, leading to increased storage costs.

Next, we explore the impact of our **context models**. As demonstrated in Figure 7 right, removing intra- and inter-Gaussian context models progressively decreases RD performance. Compared to the full FCGS model, the bit consumption is $1.5\times$ for the base model under similar fidelity conditions.

### 4.4 BOOST EXISTING COMPRESSION APPROACHES

The vanilla 3DGS (Kerbl et al., 2023) sometimes struggles with densification issues, leading to suboptimal fidelity. Pruning techniques effectively eliminate trivial Gaussians, enhancing fidelity while reducing size, particularly evident with the *Deep Blending* dataset (Hedman et al., 2018). Importantly, FCGS is compatible with these pruning techniques. We refer to Trimming (Ali et al., 2024) and Mini-Splatting (Fang & Wang, 2024), both of which apply pruning to Gaussians. As shown in Figure 8, FCGS significantly boosts their compression performance. Notably, MSC is a straightforward optimization-free compression tool utilized in Fang & Wang (2024). We compare the compression performance of FCGS and MSC by directly applying them to the same pruned 3DGS representations. The superior compression performance of FCGS underscores its significance. Please refer to Appendix Section M to find the results on the other three datasets.

### 4.5 CODING AND RENDERING EFFICIENCY ANALYSIS

Our coding time includes GPCC (Chen et al., 2023) coding for coordinates $\boldsymbol{\mu}^{\mathrm{g}}$ and arithmetic coding for attributes $\boldsymbol{f}^{\mathrm{geo}}$ and $\boldsymbol{f}^{\mathrm{col}}$ and masks $m$. On average, FCGS takes about 1 second to encode $100K$ Gaussians when running on a single GPU. Please refer to Appendix Section H to find the detailed analysis on coding time. The rendering time of the decoded 3DGS is consistent with that before compression since FCGS does not alter the number or structure of the Gaussians. For instance, the average FPS is 102 and 91 before and after compression ($\lambda = 1e - 4$) on the *MipNeRF360* dataset.

## 5 CONCLUSION

In this paper, we introduce a pioneering ***generalizable optimization-free compression*** pipeline for 3DGS representations and propose our FCGS model. FCGS enables fast compression of existing 3DGS without any finetuning, offering significant time savings. More importantly, we achieve impressive RD performance, exceeding $20\times$ compression, through the meticulous design of the MEM module and context models. Our approach can also boost compression performance of pruning-based methods. Overall, this new compression pipeline has the potential to significantly enhance the widespread application of 3DGS compression techniques due to its numerous advantages.

## ACKNOWLEDGEMENT

The paper is supported in part by The National Natural Science Foundation of China (No.62325109, U21B2013). MH is supported by funding from The Australian Research Council Discovery Program DP230101176.

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
