# – Appendix –

Table A: Table of Notation.

| Notation | Shape | Definition |
|---|---|---|
| $l$ | $\mathbb{R}^3$ | A random 3D location |
| $\boldsymbol{\mu}^{\mathrm{g}}$ | $\mathbb{R}^3$ | Location of Gaussians in 3DGS |
| $\boldsymbol{\Sigma}$ | $\mathbb{R}^{3\times3}$ | Covariance matrix of Gaussians |
| $\boldsymbol{S}$ | $\mathbb{R}^{3\times3}$ | Scale matrix of Gaussians |
| $\boldsymbol{R}$ | $\mathbb{R}^{3\times3}$ | Rotation matrix of Gaussians |
| $\alpha$ | $\mathbb{R}$ | Opacity of Gaussians after 2D projection |
| $\boldsymbol{c}$ | $\mathbb{R}^3$ | View-dependent color of Gaussians |
| $\boldsymbol{C}$ | $\mathbb{R}^3$ | The obtained pixel value after rendering |
| $\boldsymbol{f}^{\mathrm{geo}}$ | $\mathbb{R}^8$ | Geometry attribute |
| $\boldsymbol{f}^{\mathrm{col}}$ | $\mathbb{R}^{48}$ | Color attribute |
| $\boldsymbol{f}^{\mathrm{gau}}$ | $\mathbb{R}^{56}$ | Concat of $\boldsymbol{f}^{\mathrm{geo}}$ and $\boldsymbol{f}^{\mathrm{col}}$ |
| $\boldsymbol{x}$ | | Input to the autoencoder, either $\boldsymbol{f}^{\mathrm{geo}}$ or $\boldsymbol{f}^{\mathrm{col}}$ |
| $\boldsymbol{y}$ | $\mathbb{R}^{D^y}$ | Latent space of $\boldsymbol{x}$ |
| $\boldsymbol{z}$ | $\mathbb{R}^{D^z}$ | Hyperprior $\boldsymbol{y}$ |
| $\hat{\boldsymbol{x}}$ | | Decoded version of $\boldsymbol{x}$ |
| $\hat{\boldsymbol{y}}$ | $\mathbb{R}^{D^y}$ | Quantized version of $\boldsymbol{y}$ |
| $\hat{\boldsymbol{z}}$ | $\mathbb{R}^{D^z}$ | Quantized version of $\boldsymbol{z}$ |
| $m$ | $\mathbb{R}$ | The adaptive mask in MEM |
| $D^y$ | | Number of channels of $\boldsymbol{y}$ |
| $D^z$ | | Number of channels of $\boldsymbol{z}$ |
| $g_a$ | | The synthesis transform |
| $g_s$ | | The analysis transform |
| $h_a$ | | The hyper synthesis transform |
| $h_s$ | | The hyper analysis transform |
| $N^{\mathrm{g}}$ | | Number Gaussians for each 3DGS scene |
| $N^{\mathrm{s}}$ | | Number of batches for the inter-Gaussian context model |
| $N^{\mathrm{c}}$ | | Number of chunks for the intra-Gaussian context model |
| $n^{\mathrm{s}}$ | | One batch for the inter-Gaussian context model |
| $n^{\mathrm{c}}$ | | One chunk for the intra-Gaussian context model |
| $\boldsymbol{v}$ | $\mathbb{R}^3$ | A voxel of the grids |
| $\boldsymbol{f}^{\boldsymbol{v}}$ | | Feature of the voxel $\boldsymbol{v}$ |
| $w$ | | Weight for interpolation |
| $c$ | | Channel amount per chunk for the intra-Gaussian context model |
| $\boldsymbol{\mu}^{\mathrm{h}}, \boldsymbol{\sigma}^{\mathrm{h}}, \boldsymbol{\pi}^{\mathrm{h}}$ | $\mathbb{R}^{D^y}$ for each | The set of Gaussian distribution parameter from hyperprior |
| $\boldsymbol{\mu}^{\mathrm{s}}, \boldsymbol{\sigma}^{\mathrm{s}}, \boldsymbol{\pi}^{\mathrm{s}}$ | $\mathbb{R}^{D^y}$ for each | The set of Gaussian distribution parameter from inter-Gaussian context model |
| $\boldsymbol{\mu}^{\mathrm{c}}, \boldsymbol{\sigma}^{\mathrm{c}}, \boldsymbol{\pi}^{\mathrm{c}}$ | $\mathbb{R}^{D^y}$ for each | The set of Gaussian distribution parameter from intra-Gaussian context model |
| $\theta$ | | The softmax-nomalized weight for GMM mixing |
| $p$ | | Probability |
| $bit$ | | Bit consumption calculated from the probability |
| $\boldsymbol{I}$ | | The ground truth multi-view image |
| $\hat{\boldsymbol{I}}$ | | The multi-view image render from $\hat{\boldsymbol{x}}$ |
| $\Phi$ | | The cumulative Gaussian distribution function |
| $\mathbb{Y}_{[n^{\mathrm{s}}]}$ | | Set of $\hat{\boldsymbol{y}}$ for the $n^{\mathrm{c}}$-th batch |
| $\mathbb{V}$ | | Usually used in the form of $\mathbb{V}^{\boldsymbol{\mu}^{\mathrm{g}}}$, meaning voxels forming a $\boldsymbol{\mu}^{\mathrm{g}}$'s minimum bounding box. |
| $\mathbb{Q}$ | | The quantization operation |
| $q$ | | The learnable quantization step as model parameters |
| $\mathrm{Sig}$ | | The Sigmoid function |
| $\epsilon_m$ | | The hyperparameter for the binary thresholding of the mask |
| $\mathrm{emb}$ | | Sin-cos positional embedding for coordinates |
| $\oplus$ | | Concatenate operation |
| $\mathrm{MLP_m}$ | | The MLP to deduce mask in MEM |
| $\mathrm{MLP_s}$ | | The MLP to deduce Gaussian distribution parameters in inter-Gaussian context |
| $\mathrm{MLP_c}$ | | The MLP to deduce Gaussian distribution parameters in intra-Gaussian context |
| $\lambda_m$ | | The tradeoff parameter for mask rate (not used) |
| $\lambda$ | | The RD tradeoff parameter for bit and fidelity |
| $L$ | | The overall loss function |
| $L_{\mathrm{fidelity}}$ | | The fidelity loss function |
| $L_{\mathrm{entropy}}$ | | The entropy loss function |

# A  STATISTICAL DATA FOR DL3DV-GS

Using the *DL3DV* dataset (Ling et al., 2024), we generate our *DL3DV-GS* dataset through per-scene optimization with 3DGS (Kerbl et al., 2023), resulting in a total of 6770 3DGS representations. In this section, we present statistical data for the *DL3DV-GS* dataset, including its size and fidelity metrics, as exhibited below. Note that, during per-scene optimization, we adhere to the 3DGS train-test view splitting strategy, using 1 view for testing and 7 views for training across every 8 consecutive views. After optimization, we randomly select 100 scenes for testing, leaving the rest for training. All the approaches are evaluated on the test scenes. For our FCGS, it is evaluated on test views, while for other training-based methods, they are optimized (or trained) on training views and then evaluated on test views.

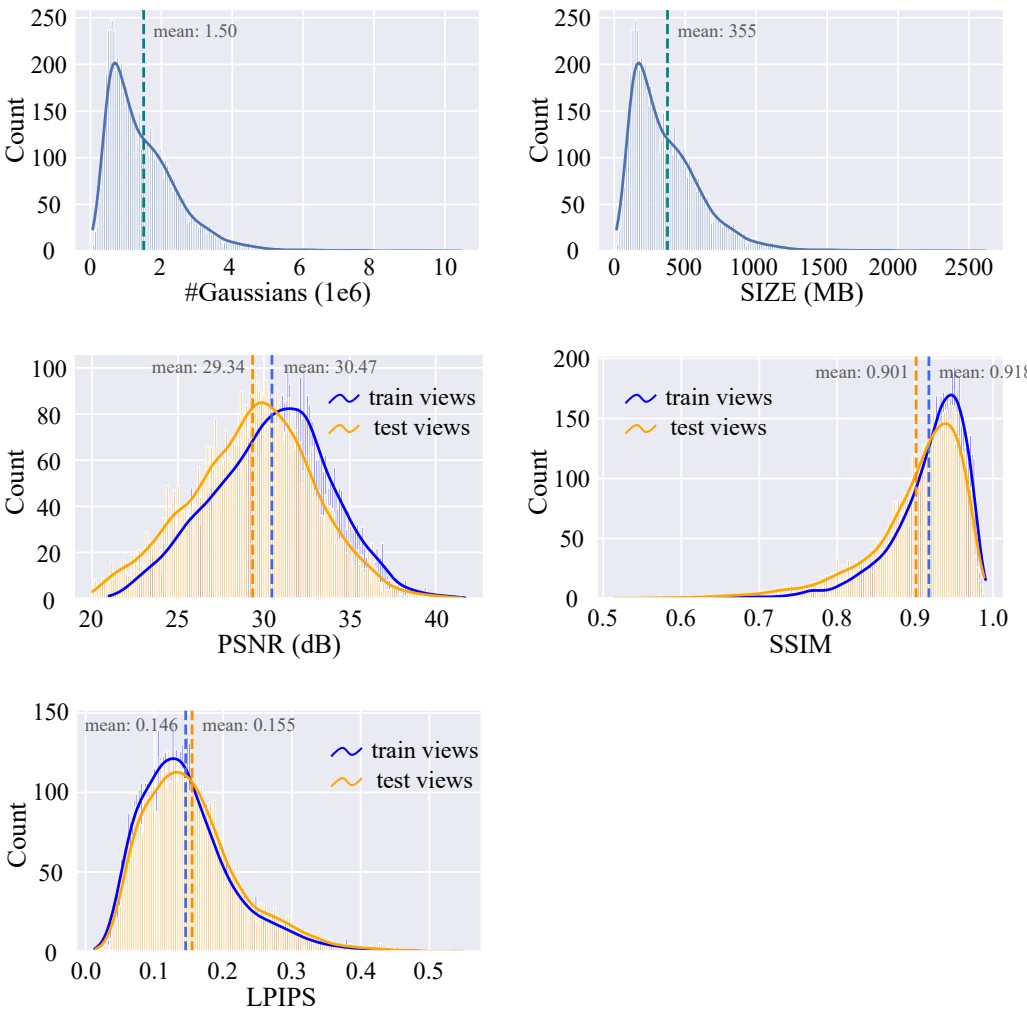

Figure A: Statistical data of the *DL3DV-GS* dataset. We also mark the mean values of each data in the sub-figures.

We also provide example images of this dataset. The images are randomly selected from test views from test scenes, which presents high fidelity.

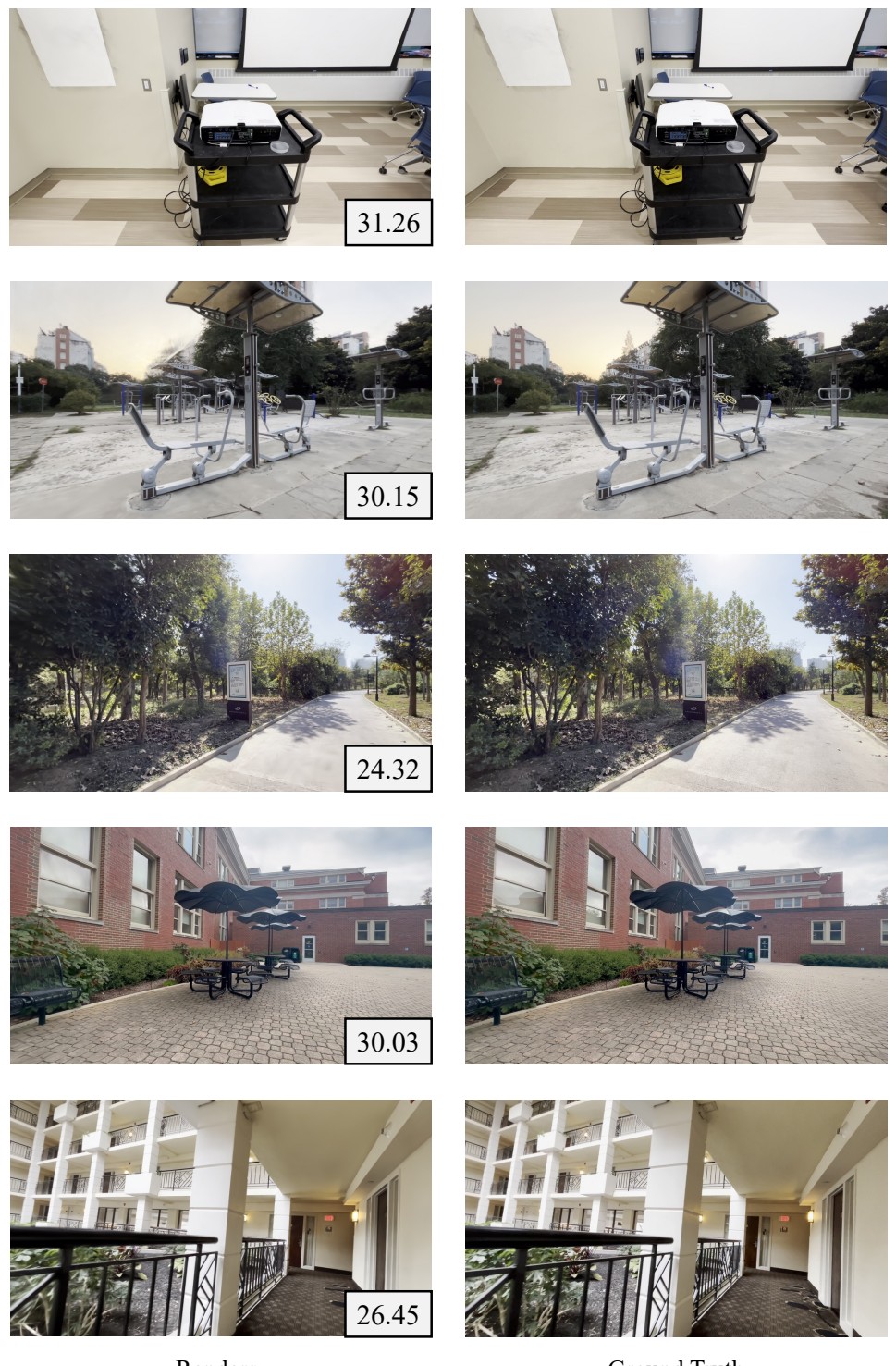

Renders                                     Ground Truth

Figure B: We randomly select several test views from test scenes for visualization, which showcase high-fidelity quality. The PSNR (dB) values for these four example scenes are indicated at the bottom-right corner of each image, which exhibit high fidelity.

## B ANALYSIS OF LIMITATIONS

To compress 3DGS from feed-forward models, we set the mask $m$ to all 0s for ***color*** attributes, instead of using MEM to adaptively infer the mask $m$. This is because the characteristics in 3DGS from feed-forward models like MVSplat (Chen et al., 2024c) differ significantly from our training data (*i.e.*, 3DGS from optimization). Specifically, **for MVSplat** (Chen et al., 2024c; Charatan et al., 2024), it uses an SH of 4 degrees (which we truncate to 3 degrees without observing significant fidelity loss), and applies a weighted mask that assigns lower weights to higher-degree SH. This results in the values of high-degree SH in its 3DGS being one order of magnitude smaller than in ours. **For LGM** (Tang et al., 2024), it uses an SH degree of 0 (*i.e.*, it only has DC coefficients). To make our model compatible with its 3DGS, we need to pad zeros for high-degree coefficients to match the shape. Due to these differences, values of these 3DGS vary greatly from ours. Failures mainly arise from incorrect selections of $m$ by MEM due to the domain gap: a Gaussian that should have been assigned to the $m = 0$ path for fidelity preservation may be mistakenly assigned to $m = 1$, and the autoencoder in this path fails to recover these Gaussian values properly from their latent space, leading to significant fidelity degradation. This issue becomes noticeable for 3DGS from feed-forward models such as MVSplat (Chen et al., 2024c), due to a distribution gap between our training data (3DGS from optimization) and those from feed-forward models like MVSplat. We mitigate this problem by enforcing all Gaussians to the $m = 0$ path for compressing 3DGS from feed-forward models, thereby prioritizing fidelity. For instance, to compress 3DGS from MVSplat, $m$ deduced from MEM may not be correct, which would result in a PSNR drop of approximately $0.7$dB at a slightly smaller compression size compared to setting all $m$ as 0s. Overall, this is our major limitation, which points out an interesting problem for future work: *how to design optimization-free compression models that can be well generalized to 3DGS with different value characteristics.*

## C EFFECT OF DIFFERENT RANDOM SEEDS ON GAUSSIAN SPLITTING

In our inter-Gaussian context models, we employ a random splitting strategy to uniformly divide all Gaussians into $N^s$ batches and decode them progressively. For decoding, we maintain the same random seed as encoding to guarantee the consistency of the contexts. In this section, we investigate the effect of different random seeds on this splitting process and its impact on the final results. Note that this only affects bit consumption but does not influence reconstruction fidelity. By testing 5 different random seeds over the test set of *DL3DV-GS* with $\lambda = 1e - 4$, we observed a standard deviation of $8e - 4$ MB, with a Coefficient of Variation (*i.e.*, standard deviation divided by the mean) of $3.6e - 5$. This experiment demonstrates the stability of our splitting approach with respect to the random seed.

We also investigated other sampling strategies like Farthest Point Sampling (FPS) in (Qi et al., 2017). Since FPS is very time-consuming for massive points in 3DGS, we evaluated only scenes with fewer than $500K$ Gaussians in the test set of *DL3DV-GS*. We observed an average bit increase of $0.6\%$. This is because FPS tends to sample points located at corners (which have the farthest distances from others), leading to non-uniform splitting and affecting the accuracy of context models.

# D    DISCUSSIONS ON SOTA 3DGS COMPRESSION METHODS

The State-of-The-Art (SoTA) 3DGS compression methods are HAC (Chen et al., 2024b) and ContextGS (Wang et al., 2024b), which utilize a ***per-scene optimization-based compression*** pipeline and exhibit better compression performance over our FCGS. However, since we configure FCGS without per-scene optimization, this setup naturally puts it at a disadvantage when compared with optimization-based methods, and indeed, there is a performance gap compared to these two methods (*i.e.*, HAC and ContextGS). However, these two approaches employ anchor-based structures (Lu et al., 2024), whose representations diverge from the standard 3DGS structure and are inherently about $5\times$ smaller than the vanilla 3DGS. In this paper, we target FCGS at the standard 3DGS structure, same as the comparative methods in our experiment. It is also promising to extend FCGS to anchor-based 3DGS variants (Lu et al., 2024) to achieve better RD performance, which we leave for future work.

# E    MASK RATIO ANALYSIS IN MEM

We provide statistical data on the ratio of path $m = 1$ in MEM. As $\lambda$ increases, the model is expected to achieve lower bitrates at the cost of reduced fidelity. To accomplish this, the model tends to assign more 1s to the mask $m$, thereby allowing more ***color*** attributes $\boldsymbol{f}^{\text{col}}$ to eliminate redundancies via the autoencoder (*i.e.*, the path of $m = 1$), and vice versa.

Table B: Ratio of path $m = 1$ in MEM.

| $\lambda$ | DL3DV-GS | MipNeRF360 | Tank&Temples |
|---|---|---|---|
| $1e-4$ | 0.62 | 0.48 | 0.60 |
| $2e-4$ | 0.67 | 0.55 | 0.66 |
| $4e-4$ | 0.74 | 0.63 | 0.74 |
| $8e-4$ | 0.82 | 0.74 | 0.82 |
| $16e-4$ | 0.89 | 0.85 | 0.90 |

# F MORE FIDELITY METRICS AND TRAINING TIME

We provide additional fidelity metrics, including PSNR, SSIM, and LPIPS. We also report the training time for each method. For those marked with *, it represents the time taken for finetuning from a common existing 3DGS. For those marked with **, it represents the time taken for training from scratch. For our method, it is the encoding time using multiple/single GPUs.

Table C: Experiments on *DL3DV-GS* dataset. Methods marked with * are finetuned from a common 3DGS (our FCGS compresses the same 3DGS). Methods marked with ** are trained from scratch. The notation "(751 +)" indicates the training time of the vanilla 3DGS.

| Methods | PSNR (dB) ↑ | SSIM ↑ | LPIPS ↓ | SIZE (MB) ↓ | TIME (s) ↓ |
|---|---|---|---|---|---|
| 3DGS | 29.34 | 0.898 | 0.146 | 372.50 | 751 |
| Light* | 28.89 | 0.890 | 0.159 | 24.61 | (751 +) 227 |
| Navaneet* | 28.36 | 0.884 | 0.169 | 13.60 | (751 +) 546 |
| Simon* | 29.04 | 0.893 | 0.154 | 15.36 | (751 +) 122 |
| SOG** | 28.54 | 0.888 | 0.156 | 16.50 | 1068 |
| EAGLES** | 28.53 | 0.884 | 0.170 | 24.04 | 518 |
| Lee** | 28.71 | 0.886 | 0.165 | 36.56 | 938 |
| Ours-lowrate | 28.86 | 0.891 | 0.156 | 15.83 | (751 +) 9 / 16 |
| Ours-highrate | 29.26 | 0.897 | 0.148 | 27.44 | (751 +) 11 / 20 |

Table D: Experiments on *MipNeRF360* dataset. Methods marked with * are finetuned from a common 3DGS (our FCGS compresses the same 3DGS). Methods marked with ** are trained from scratch. The notation "(1583 +)" indicates the training time of the vanilla 3DGS.

| Methods | PSNR (dB) ↑ | SSIM ↑ | LPIPS ↓ | SIZE (MB) ↓ | TIME (s) ↓ |
|---|---|---|---|---|---|
| 3DGS | 27.52 | 0.813 | 0.221 | 741.12 | 1583 |
| Light* | 27.31 | 0.808 | 0.235 | 48.61 | (1583 +) 422 |
| Navaneet* | 26.80 | 0.796 | 0.256 | 20.66 | (1583 +) 973 |
| Simon* | 27.15 | 0.802 | 0.242 | 27.71 | (1583 +) 195 |
| SOG** | 27.08 | 0.799 | 0.229 | 41.00 | 2391 |
| EAGLES** | 27.14 | 0.809 | 0.231 | 58.91 | 1245 |
| Lee** | 27.05 | 0.797 | 0.247 | 48.93 | 1885 |
| Ours-lowrate | 27.05 | 0.798 | 0.237 | 34.64 | (1583 +) 10 / 31 |
| Ours-highrate | 27.39 | 0.806 | 0.226 | 64.05 | (1583 +) 14 / 41 |

Table E: Experiments on *Tank&Temples* dataset. Methods marked with * are finetuned from a common 3DGS (our FCGS compresses the same 3DGS). Methods marked with ** are trained from scratch. The notation "(814 +)" indicates the training time of the vanilla 3DGS.

| Methods | PSNR (dB) ↑ | SSIM ↑ | LPIPS ↓ | SIZE (MB) ↓ | TIME (s) ↓ |
|---|---|---|---|---|---|
| 3DGS | 23.71 | 0.845 | 0.179 | 432.03 | 814 |
| Light* | 23.62 | 0.837 | 0.197 | 28.60 | (814 +) 241 |
| Navaneet* | 23.15 | 0.831 | 0.205 | 13.64 | (814 +) 587 |
| Simon* | 23.63 | 0.842 | 0.187 | 17.65 | (814 +) 120 |
| SOG** | 23.66 | 0.837 | 0.187 | 23.10 | 1219 |
| EAGLES** | 23.28 | 0.835 | 0.203 | 28.99 | 604 |
| Lee** | 23.35 | 0.832 | 0.202 | 39.38 | 1007 |
| Ours-lowrate | 23.48 | 0.832 | 0.193 | 17.89 | (814 +) 10 / 16 |
| Ours-highrate | 23.62 | 0.839 | 0.184 | 32.02 | (814 +) 13 / 24 |

# G  ANALYSIS OF STORAGE SIZE OF DIFFERENT COMPONENTS

We provide the storage size of each component from the *DL3DV-GS* dataset, as shown in the tables below. These two tables exhibit the total storage size (in MB) and the per-parameter size (in bit) of each component, respectively. For the coordinates $\boldsymbol{\mu}^{\mathrm{g}}$, they are consistently compressed using the same GPCC command across different $\lambda$ values, resulting in a same size. For other components, as $\lambda$ decreases, the storage required for both geometry attributes $\boldsymbol{f}^{\mathrm{geo}}$ and color attributes $\boldsymbol{f}^{\mathrm{col}}$ ($m = 1$ plus $m = 0$) reduces. Focusing on the color attributes $\boldsymbol{f}^{\mathrm{col}}$, the mask rate increases as $\lambda$ becomes larger, shifting more $\boldsymbol{f}^{\mathrm{col}}$ to the $m = 1$ path for redundancy elimination via the autoencoder structure. With a higher portion of $\boldsymbol{f}^{\mathrm{col}}$ going through the $m = 1$ path, the overall storage of this path may increase (while the bits per parameter still reduce). The storage for the $m = 0$ path decreases significantly due to simultaneously reduced selection ratios of this path and decreased per-parameter size under stronger entropy constraints. The mask size itself also reduces as the mask rate increases, since in a binary distribution, greater dominance of one value (*i.e.*, 1 in our case) lowers the entropy.

Regarding the challenges in compression, Table G shows that $\boldsymbol{f}^{\mathrm{col}}$ ($m = 1$) is the easiest to compress, as it is compressed in a latent space using autoencoder. In contrast, $\boldsymbol{f}^{\mathrm{geo}}$ is more difficult to compress due to the complexity and sensitivity of geometric attributes, which have greater variability and less mutual information. The compression ratio could be calculated as 32 divided by a bit value in Table G, since the original parameters (*i.e.*, those before compression) are stored in `float32`, which allocates 32 bits for each parameter.

Table F: **Storage size** of different components on the *DL3DV-GS* dataset. All sizes are measured in **MB**. The mask rate indicates the proportion of color attributes $\boldsymbol{f}^{\mathrm{col}}$ assigned to the $m = 1$ path (*i.e.*, compressed via the autoencoder).

| $\lambda$ | Total size | $\boldsymbol{\mu}^{\mathrm{g}}$ | $\boldsymbol{f}^{\mathrm{col}}$ ($m$=1) | $\boldsymbol{f}^{\mathrm{col}}$ ($m$=0) | $\boldsymbol{f}^{\mathrm{geo}}$ | Mask size | Mask rate |
|---|---|---|---|---|---|---|---|
| $1e-4$ | 27.44 | 3.25 | 2.61 | 10.92 | 10.50 | 0.17 | 0.62 |
| $2e-4$ | 24.15 | 3.25 | 2.72 | 8.10 | 9.93 | 0.16 | 0.67 |
| $4e-4$ | 21.12 | 3.25 | 2.77 | 5.67 | 9.29 | 0.14 | 0.74 |
| $8e-4$ | 18.14 | 3.25 | 3.01 | 3.42 | 8.34 | 0.12 | 0.82 |
| $16e-4$ | 15.83 | 3.25 | 2.94 | 1.76 | 7.80 | 0.08 | 0.89 |

Table G: **Per-parameter bits** of different components on the *DL3DV-GS* dataset. All sizes are measured in **bit**. The mask rate indicates the proportion of color attributes $\boldsymbol{f}^{\mathrm{col}}$ assigned to the $m = 1$ path (*i.e.*, compressed via the autoencoder).

| $\lambda$ | Weighted AVG | $\boldsymbol{\mu}^{\mathrm{g}}$ | $\boldsymbol{f}^{\mathrm{col}}$ ($m$=1) | $\boldsymbol{f}^{\mathrm{col}}$ ($m$=0) | $\boldsymbol{f}^{\mathrm{geo}}$ | Mask size | Mask rate |
|---|---|---|---|---|---|---|---|
| $1e-4$ | 2.51 | 5.92 | 0.45 | 3.29 | 6.94 | 0.94 | 0.62 |
| $2e-4$ | 2.21 | 5.92 | 0.43 | 2.92 | 6.56 | 0.89 | 0.67 |
| $4e-4$ | 1.94 | 5.92 | 0.40 | 2.58 | 6.14 | 0.81 | 0.74 |
| $8e-4$ | 1.67 | 5.92 | 0.39 | 2.32 | 5.51 | 0.66 | 0.82 |
| $16e-4$ | 1.45 | 5.92 | 0.36 | 2.10 | 5.15 | 0.48 | 0.89 |

# H ANALYSIS OF ENCODING/DECODING TIME OF DIFFERENT COMPONENTS

We provide the encoding/decoding time for each component from the *DL3DV-GS* dataset, as shown in the tables below. For the coordinates $\boldsymbol{\mu}^{\mathrm{g}}$, the same GPCC command is applied consistently across different $\lambda$ values, resulting in a similar time consumption. Other components (*i.e.*, $\boldsymbol{f}^{\mathrm{col}}$, $\boldsymbol{f}^{\mathrm{geo}}$ and the mask) are encoded/decoded using arithmetic codec. As $\lambda$ increases, entropy decreases, leading to reduced codec time. With a higher $\lambda$, an increased mask rate shifts more $\boldsymbol{f}^{\mathrm{col}}$ to the $m = 1$ path, which raises the time for this path (though the time per parameter still reduces considering the increased mask rate). The time for the $m = 0$ path decreases significantly due to simultaneously reduced selection ratios of this path and decreased per-parameter time under stronger entropy constraints. The "Others" time includes network forward passes and supplementary operations. With bitrates decreases (*i.e.*, with higher $\lambda$), more $\boldsymbol{f}^{\mathrm{col}}$ is assigned to the $m = 1$ path, which involves an autoencoder. This leads to a slight increase in network forward time which is categorized under "Others". Notably, during encoding, most time is spent on GPCC, while arithmetic coding remains efficient. During decoding, the arithmetic decoder's index search raises decoding complexity over encoding, while GPCC decoding is faster than encoding since it does not require RD search. Overall, decoding is faster than encoding.

Table H: **Encoding time** of different components on the *DL3DV-GS* dataset, which has over 1.57 million Gaussians on average in the testing set. All times are measured in seconds, with each component's percentage share of the total encoding time provided in parentheses. The mask rate indicates the proportion of color attributes $\boldsymbol{f}^{\mathrm{col}}$ assigned to the $m = 1$ path (*i.e.*, compressed via the autoencoder). Results are obtained using a single GPU.

| $\lambda$ | Total time | $\boldsymbol{\mu}^{\mathrm{g}}$ | $\boldsymbol{f}^{\mathrm{col}}$ ($m$=1) | $\boldsymbol{f}^{\mathrm{col}}$ ($m$=0) | $\boldsymbol{f}^{\mathrm{geo}}$ | Mask | Others | Mask rate |
|---|---|---|---|---|---|---|---|---|
| $1e-4$ | 20.47 | 9.48 (46%) | 2.81 (14%) | 4.41 (22%) | 2.35 (11%) | 0.02 (0%) | 1.41 (7%) | 0.62 |
| $2e-4$ | 18.33 | 9.46 (52%) | 2.97 (16%) | 2.63 (14%) | 1.83 (10%) | 0.02 (0%) | 1.42 (8%) | 0.67 |
| $4e-4$ | 17.17 | 9.50 (55%) | 3.09 (18%) | 1.67 (10%) | 1.46 (8%) | 0.02 (0%) | 1.44 (8%) | 0.74 |
| $8e-4$ | 16.44 | 9.49 (58%) | 3.34 (20%) | 1.01 (6%) | 1.10 (7%) | 0.02 (0%) | 1.48 (9%) | 0.82 |
| $16e-4$ | 15.86 | 9.44 (60%) | 3.21 (20%) | 0.68 (4%) | 1.00 (6%) | 0.02 (0%) | 1.51 (10%) | 0.89 |

Table I: **Decoding time** of different components on the *DL3DV-GS* dataset, which has over 1.57 million Gaussians on average in the testing set. All times are measured in seconds, with each component's percentage share of the total decoding time provided in parentheses. The mask rate indicates the proportion of color attributes $\boldsymbol{f}^{\mathrm{col}}$ assigned to the $m = 1$ path (*i.e.*, compressed via the autoencoder). Results are obtained using a single GPU.

| $\lambda$ | Total time | $\boldsymbol{\mu}^{\mathrm{g}}$ | $\boldsymbol{f}^{\mathrm{col}}$ ($m$=1) | $\boldsymbol{f}^{\mathrm{col}}$ ($m$=0) | $\boldsymbol{f}^{\mathrm{geo}}$ | Mask | Others | Mask rate |
|---|---|---|---|---|---|---|---|---|
| $1e-4$ | 15.93 | 3.53 (22%) | 3.85 (24%) | 5.07 (32%) | 2.51 (16%) | 0.02 (0%) | 0.95 (6%) | 0.62 |
| $2e-4$ | 14.16 | 3.52 (25%) | 4.21 (30%) | 3.28 (23%) | 2.14 (15%) | 0.02 (0%) | 0.99 (7%) | 0.67 |
| $4e-4$ | 12.84 | 3.52 (27%) | 4.33 (34%) | 2.16 (17%) | 1.77 (14%) | 0.02 (0%) | 1.04 (8%) | 0.74 |
| $8e-4$ | 12.13 | 3.53 (29%) | 4.68 (39%) | 1.40 (12%) | 1.39 (11%) | 0.02 (0%) | 1.10 (9%) | 0.82 |
| $16e-4$ | 11.47 | 3.52 (31%) | 4.67 (41%) | 0.87 (8%) | 1.24 (11%) | 0.02 (0%) | 1.15 (10%) | 0.89 |

# I ABLATION ON THE NUMBER OF SPLITS IN CONTEXT MODELS

We conduct further ablation studies on the number of splits in context models, as summarized below.

- **Inter-Gaussian context models.** In Table J, increasing the number of splits leads to size reduction with the same fidelity under the given digital precision. On the one hand, testing with only 2 splits using an extreme ratio of $95\%, 5\%$ demonstrates poor performance, underscoring the importance of rational split strategies. After that, the most notable improvement occurs when increasing from 3 splits to 4 splits, achieving a size reduction of $0.227$ MB. Beyond 4 splits, the size reduction becomes negligible. On the other hand, even in the extreme 2-split case, the size remains significantly smaller than in the ablation study shown in Figure 7, where the inter-Gaussian context model is entirely removed. This is because, while $\boldsymbol{f}^{\boldsymbol{\mu}^{\mathrm{g}}}$ is unavailable for the first batch (we manually set it to 0 as a placeholder) in Equation 5, the positional encoding $\mathrm{emb}(\boldsymbol{\mu}^{\mathrm{g}})$ still provides effective context.

- **Intra-Gaussian context models.** In Table K, increasing the number of chunks does not consistently yield size reduction. For 2 chunks, insufficient context information leads to a size increase. Expanding to 8 or 16 chunks does not always provide further benefits due to the increased complexity of the model, which makes convergence more challenging.

Table J: Ablation study on the number of splits in the inter-Gaussian context model on the *DL3DV-GS* dataset with $\lambda = 1e-4$. We change inter-split number of all $\boldsymbol{f}^{\mathrm{col}}$ ($m = 1$), $\boldsymbol{f}^{\mathrm{col}}$ ($m = 0$), and $\boldsymbol{f}^{\mathrm{geo}}$. Our FCGS employs a default setting of $17\%, 17\%, 33\%, 33\%$ (4 splits).

| # Split Batches | SIZE (MB) ↓ | PSNR (dB) ↑ | SSIM ↑ | LPIPS ↓ |
|---|---|---|---|---|
| $95\%, 5\%$ (2) | 28.685 | 29.264 | 0.897 | 0.148 |
| $25\%, 25\%, 50\%$ (3) | 27.667 | 29.264 | 0.897 | 0.148 |
| $17\%, 17\%, 33\%, 33\%$ (4) | 27.440 | 29.264 | 0.897 | 0.148 |
| $13\%, 13\%, 25\%, 25\%, 25\%$ (5) | 27.379 | 29.264 | 0.897 | 0.148 |

Table K: Ablation study on the number of splits in the intra-Gaussian context model on the *DL3DV-GS* dataset with $\lambda = 1e-4$. We change intra-split number of $\boldsymbol{f}^{\mathrm{col}}$ ($m = 1$). Our FCGS employs a default setting of 4 chunks.

| # Split Chunks | SIZE (MB) ↓ | PSNR (dB) ↑ | SSIM ↑ | LPIPS ↓ |
|---|---|---|---|---|
| 2 | 28.562 | 29.267 | 0.897 | 0.148 |
| 4 | 27.440 | 29.264 | 0.897 | 0.148 |
| 8 | 27.516 | 29.264 | 0.897 | 0.148 |
| 16 | 27.649 | 29.265 | 0.897 | 0.148 |

## J    VISUALIZATION OF BIT ALLOCATION USING CONTEXT MODELS

We present a visualization of bits per parameter using the proposed inter- and intra-Gaussian context models, shown in the figure below.

- **For the inter-Gaussian context model**: From a statistical perspective, each Gaussian in different batches is expected to hold similar amounts of information on average due to the random splitting strategy, which ensures an even distribution of Gaussians across the 3D space. Thus, as batches progress deeper (*i.e.*, B1 → B4), the bits per parameter decrease. The most significant reduction occurs between B1 and B2, since B1 lacks inter-Gaussian context, leading to less accurate probability estimates. For subsequent batches, the reduction is less pronounced, as B1 already includes $1/6$ of the total Gaussians, providing sufficient context for accurate prediction. Adding more Gaussians from later batches yields diminishing benefits. This finding validates our approach of splitting batches with varying proportions of Gaussians, where the first two batches contain fewer Gaussians. **For the intra-Gaussian context model**: In (a), different from that in the inter-Gaussian context model, the information distribution among different chunks is not guaranteed to be equal, as the latent space is derived from a learnable MLP. As a result, C1 receives the least information because it lacks intra-Gaussian context, leading to less accurate probability predictions. Therefore, allocating more information to C1 would make entropy reduction challenging. To counter this, the neural network compensates by assigning minimal information to C1, thereby saving bits. Conversely, C2 receives the most information as it plays a pivotal role in predicting subsequent chunks (*i.e.*, C3 and C4), while its own entropy can be reduced by leveraging C1. As a result, the bits per parameter decrease consistently from C2 to C4 due to the increasingly rich context. In (b), the bit variation reflects differences among color components, where Green has the least information and is the easiest to predict.

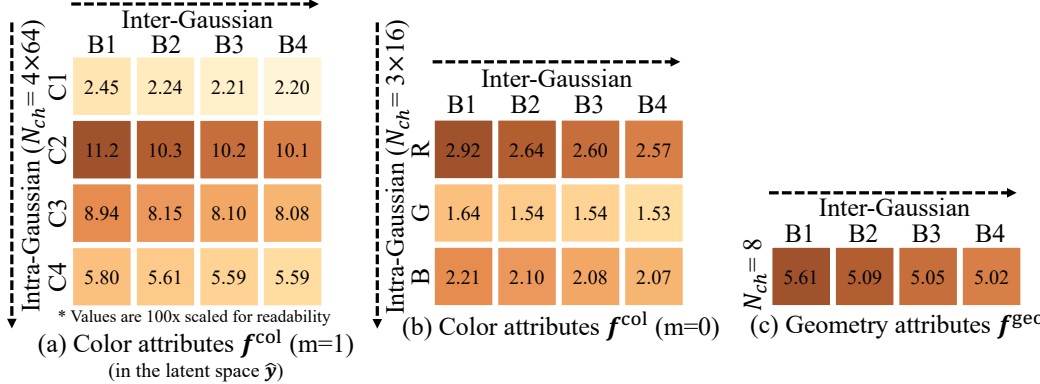

Figure C: Visualization of bits per parameter on the *DL3DV-GS* dataset with $\lambda = 16e-4$. B1 → B4 and C1 → C4 indicate batches and chunks of inter- and intra-Gaussian context models, respectively. Bit allocation for both color and geometry attributes using inter- and intra-Gaussian context models is shown. Note that in (a), the bits per parameter for color attributes $\boldsymbol{f}^{\text{col}}$ ($m = 1$) are calculated in the latent space (*i.e.*, $\hat{y}$) as $\frac{\text{per chunk bits in } \hat{y}}{64}$ for each chunk, where 64 is the number of channels per chunk of $\hat{y}$. This calculation is used for bit allocation visualization of each chunk in $\hat{y}$. This differs slightly from Table G, which takes a holistic view and calculates bits per parameter as $\frac{\text{total bits in } \hat{y}}{48}$, where 48 is the total number of channels for $\boldsymbol{f}^{\text{col}}$ ($m = 1$). The bit values in (a) are scaled by 100x for readability.

# K ABLATION STUDIES ON THE HYPERPRIOR

We present statistical data for ablation studies, and also conduct ablation experiment over the hyperprior, as shown in Table M. This hyperprior follows the approach of Ballé et al. (2018), which is originally designed for image compression, and we consistently retain it in our FCGS model. In this section, we investigate the effect of the hyperprior by manually setting $\hat{z}$ to all zeros to remove the information contributed from the hyperprior, while still preserving a necessary Gaussian probability for entropy estimation (which is now predicted from an all-zero hyperprior $\hat{z}$ instead).

Interestingly, our experiments reveal the hyperprior does not significantly contribute to the compression performance. This suggests there is less redundancy within each Gaussian attribute in 3DGS compared to that within each individual image Ballé et al. (2018). Instead, our inter- and intra-context models, which are for the unique characteristics of 3DGS, have shown more effectiveness.

Table L: Statistical data from ablation studies over MEM on the *DL3DV-GS* dataset. Data are presented as PSNR (dB) / SIZE (MB).

| $\lambda$ | $1e-4$ | $2e-4$ | $4e-4$ | $8e-4$ | $16e-4$ |
|---|---|---|---|---|---|
| Ours | 29.26 / 27.44 | 29.22 / 24.15 | 29.17 / 21.12 | 29.05 / 18.14 | 28.86 / 15.83 |
| Ours w $m$ all 0s | 29.27 / 38.15 | 29.23 / 33.87 | 29.16 / 30.12 | 28.98 / 25.99 | 28.69 / 22.84 |
| Ours w $m$ all 1s | 28.82 / – | | | | |

Table M: Statistical data from ablation studies over context models on the *DL3DV-GS* dataset. Data are presented as PSNR (dB) / SIZE (MB).

| $\lambda$ | $1e-4$ | $2e-4$ | $4e-4$ | $8e-4$ | $16e-4$ |
|---|---|---|---|---|---|
| Ours | 29.26 / 27.44 | 29.22 / 24.15 | 29.17 / 21.12 | 29.05 / 18.14 | 28.86 / 15.83 |
| Ours w/o intra | 29.26 / 29.65 | 29.22 / 25.94 | 29.15 / 23.32 | 29.02 / 18.87 | 28.82 / 16.80 |
| Ours w/o intra & inter | 29.26 / 38.55 | 29.22 / 35.63 | 29.14 / 31.15 | 29.02 / 26.50 | 28.81 / 23.53 |
| Ours w/o intra & inter & hyper | 29.25 / 39.41 | 29.22 / 34.79 | 29.17 / 31.29 | 29.05 / 27.47 | 28.84 / 24.66 |

# L TRAINING WITH LESS DATA

As outlined in the implementation, we train FCGS on 6670 3DGS scenes generated from the DL3DV dataset (Ling et al., 2024). Although preparing this training data is time-intensive, it is a one-time investment: Once trained, FCGS can be directly applied for compression without requiring further optimization, making the effort highly valuable. To evaluate FCGS's performance trained on significantly less data, we conducted experiments using only 100 scenes, as shown in Table N. Even with this reduced training set, FCGS still delivers strong results, achieving comparable fidelity metrics with only 12.9% increase in size. This robustness is due to FCGS's compact and efficient architecture, with a total model size of less than 10 MB. This finding suggests a promising direction for future work: training FCGS with a larger and more diverse dataset to further enhance its capability.

Table N: Results of **training with less data**. Experiments are conducted on the *DL3DV-GS* dataset with $\lambda = 1e-4$.

| # Trainig data | SIZE (MB) $\downarrow$ | PSNR (dB) $\uparrow$ | SSIM $\uparrow$ | LPIPS $\downarrow$ |
|---|---|---|---|---|
| Full (# 6670 scenes) | 27.440 | 29.264 | 0.897 | 0.148 |
| Subset (# 100 scenes) | 30.981 | 29.256 | 0.897 | 0.148 |

## M  BOOST COMPRESSION PERFORMANCE OF PRUNING-BASED APPROACHES

Mini-Splatting (Fang & Wang, 2024) and Trimming (Ali et al., 2024) effectively prune trivial Gaussians, thereby slimming the 3DGS. By applying our FCGS on top of their pruned 3DGS, we can achieve superior compression performance. We further evaluate this scheme across *DL3DV-GS*, *MipNeRF360*, and *Tank&Temples* datasets.

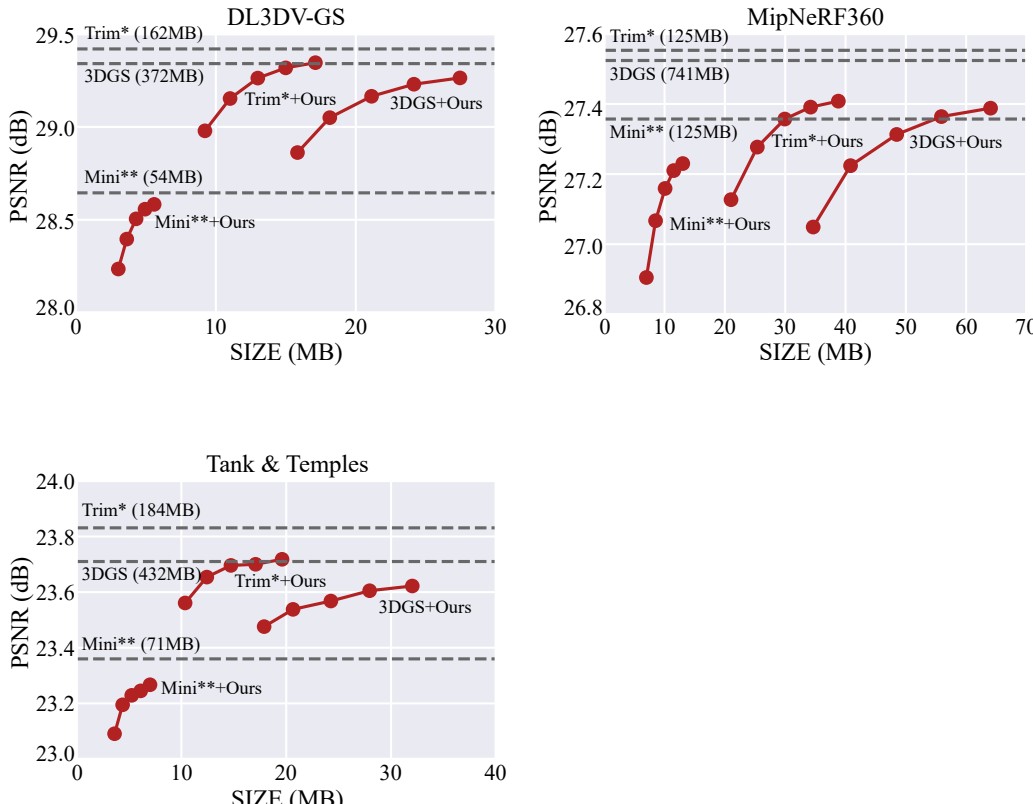

Figure D: Our FCGS boosts the compression performance of pruning-based approaches. Built on Mini-Splatting (Fang & Wang, 2024) and Trimming (Ali et al., 2024), we achieve superior RD performance by *directly* applying FCGS to their pruned 3DGS, without any finetuning.

# N ADDITIONAL QUALITATIVE COMPARISONS

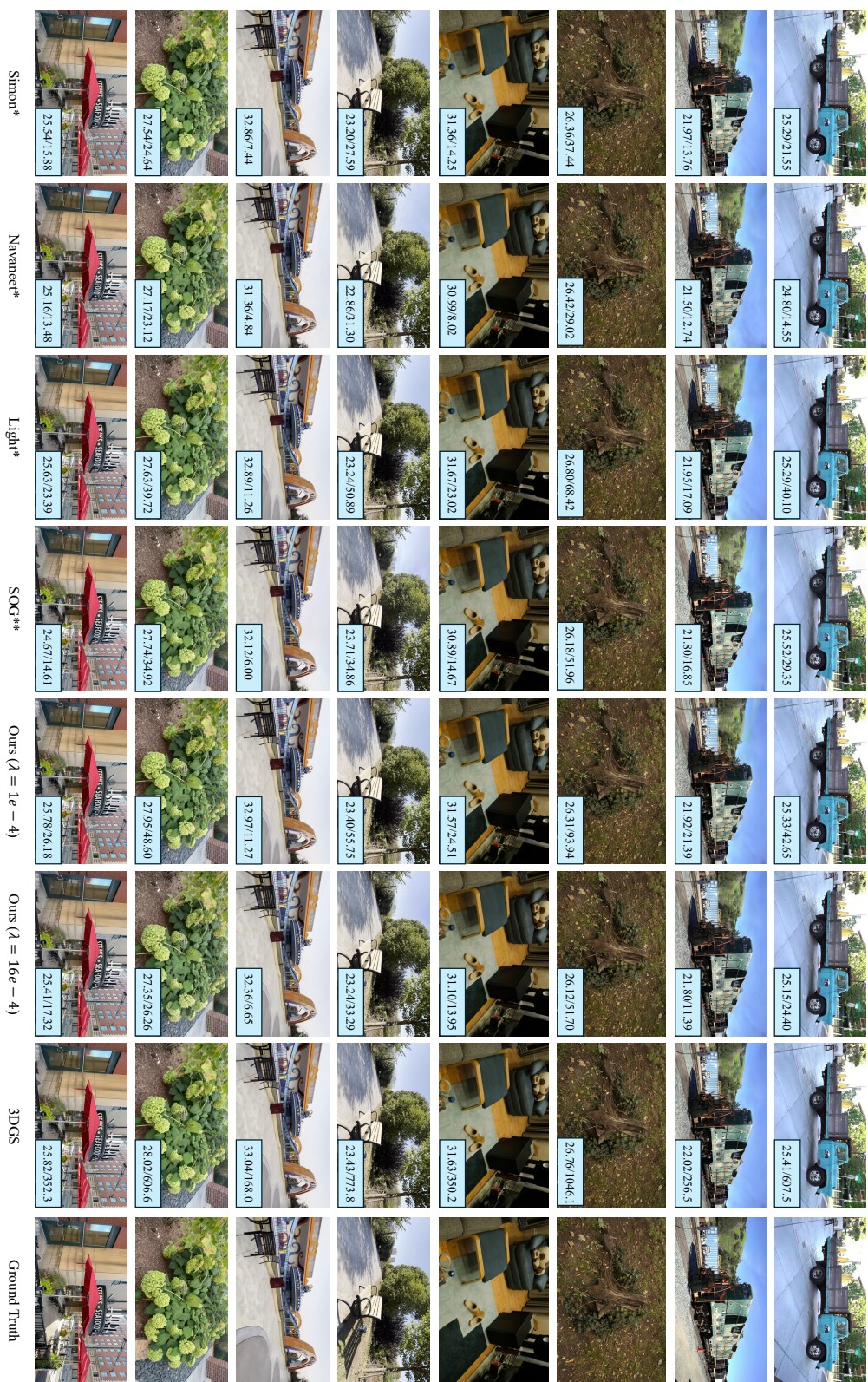

Figure E: Qualitative comparison with baselines methods. Zoom in for more details. We achieve substantial size reduction while preserving high fidelity. PSNR (dB) / SIZE (MB) are indicated in the bottom-right corner of each image.