# OpenReview forum: "Fast Feedforward 3D Gaussian Splatting Compression"
_ICLR.cc/2025/Conference — ICLR 2025 Poster_

### Official Review · Reviewer_SkMb · 2024-10-23

**Soundness:** 3
**Presentation:** 2
**Contribution:** 3
**Rating:** 6
**Confidence:** 4

**Summary:**

This paper introduces an optimization-free model, called FCGS, for 3D Gaussian Splatting (3DGS) compression, which allows compressing an optimized 3DGS representation in a feed-forward process. The model incorporates both intra- and inter-Gaussian context information. The authors project most Gaussian attributes into a latent space and utilize arithmetic encoding and decoding for data compression. Additionally, they employ an end-to-end progressive training scheme for the proposed pipeline.

**Strengths:**

As far as I know, this is the first generalizable optimization-free 3DGS compression pipeline. Without the need of  original  training images, this generalizable compression method enables the compression of 3DGS, expanding the application scope of 3DGS compression algorithms.

This article provides a novel approach to context modeling for exploring the correlations between Gaussians.

**Weaknesses:**

I find the experimental content of this paper somewhat disorganized and difficult to follow. The authors present the main comparative experimental results using figures, but there are no numerical comparisons in the main paper. Although detailed data is provided in the Appendix, the absence of tables in the main paper causes some inconvenience for readers.

Additionally, in Tables C-D, the authors report the running time, but I believe this comparison may not be entirely reasonable. The running time for methods that involve training from scratch includes the time for reconstruction. FCGS also requires a pre-trained 3DGS, and the time spent pre-training the 3DGS should be reflected in the table.

In the comparative experiments, the authors only compare methods with worse rendering quality than FCGS. However, there are existing 3DGS compression methods that offer better rendering quality and higher compression rates than FCGS. While these methods are not generalizable, readers should still be informed of the current gap between FCGS and the state-of-the-art.

**Questions:**

It seems that the process of generating m by MLP_m is not illustrated in Figure 2. By reading the method section, it is somewhat difficult for readers to understand the role and position of m in the pipeline (although there is a formula provided). It is suggested to draw a more detailed pathway in Figure 2.

---

> ### Author Response · Authors · 2024-11-22
> **Response to Reviewer 4 (SkMb)**
>
> **W1: Figure for performance comparison.**
>
> Thanks for pointing this out. Using curves to compare compression performance (*i.e.*, RD performance) is a widely adopted practice in the compression field [1][2]. Curves provide an intuitive visualization of the relative performance variances among different compression approaches, where a curve closer to the upper-left indicates better performance. Meanwhile, Table C to Table E in **Appendix Section F** serves as a necessary supplement, presenting precise numerical data for a detailed comparison. In future versions, we will strive to incorporate more quantitative data into the main paper without significantly changing the current content.
>
> **W2: Running time of 3DGS.**
>
> Thanks for your comments. We have now added a parenthesis indicating the training time of 3DGS in front of the time consumption of finetuning-based approaches and our approach in Tables C to Table E.
>
> **W3: Comparison with SoTA methods.**
>
> Since we configure FCGS without per-scene optimization, this setup naturally puts it at a disadvantage when compared with optimization-based methods, and indeed, there is a performance gap compared to SoTA methods like HAC [3] and ContextGS [4]. However, these two approaches employ anchor-based structures [5], whose representations diverge from the standard 3DGS structure and are inherently about 5X smaller than the vanilla 3DGS. In this paper, we target FCGS at the standard 3DGS structure, same as the comparative methods in our experiment. It is also promising to extend FCGS to anchor-based 3DGS variants [5] to achieve better RD performance, which we leave for future work. We have added this discussion to **Appendix Section D** and pointed out the SoTA of per-scene optimization-based 3DGS compression methods.
>
> **Q1: Regarding $MLP_m$ and Figure 2.**
>
> Thanks for pointing this out.
>
> - We have now re-formulated Equation (3) to clarify the process of generating $m$ using $MLP_m$ and updated it in the paper. The updated formulation is as follows: $$m_i = \text{Sig}(MLP_m(f^\text{gau}_i)) > \epsilon_m, $$where $\epsilon_m$ is a hyperparameter that defines the threshold for binarizing the mask, and gradients of the binarized mask $m$ are backpropagated using STE [6][7].
> - We have also improved Figure 2 to show the process of generating $m$ by $MLP_m$, and also make the pathway of the mask $m$ easier to understand.
>
> Please find them in our revised paper.
>
>
> **If you have any further questions or concerns, please feel free to reach out to us for discussion.**
>
>
> **Reference:**
>
> [1] Ballé J, Minnen D, Singh S, et al. Variational image compression with a scale hyperprior [C]. *ICLR 2018*.
>
> [2] He D, Zheng Y, Sun B, et al. Checkerboard context model for efficient learned image compression [C]. *CVPR 2021*.
>
> [3] Chen Y, Wu Q, Lin W, et al. HAC: Hash-grid Assisted Context for 3D Gaussian Splatting Compression [C]. *ECCV 2024*.
>
> [4] Wang Y, Li Z, Guo L, et al. ContextGS: Compact 3D Gaussian Splatting with Anchor Level Context Model [C]. *NIPS 2024*.
>
> [5] Lu T, Yu M, Xu L, et al. Scaffold-GS: Structured 3D Gaussians for View-Adaptive Rendering [C]. *CVPR 2024*.
>
> [6] Bengio Y, Léonard N, Courville A. Estimating or propagating gradients through stochastic neurons for conditional computation[J]. *arXiv preprint, 2013*.
>
> [7] Lee J C, Rho D, Sun X, et al. Compact 3d gaussian representation for radiance field[C]. *CVPR 2024*.

---

> ### Comment · Reviewer_SkMb · 2024-11-26
>
> Thanks for your response.  This has addressed most of my concerns. I will maintain my positive rating.

---

> > ### Author Response · Authors · 2024-11-27
> > **Response to Reviewer 4 (SkMb)**
> >
> > Thank you for your thoughtful engagement and valuable suggestions, which have been instrumental in improving our paper. We sincerely appreciate your support and the time you dedicated to reviewing our work! Please feel free to reach out if you have any further concerns.

---

### Official Review · Reviewer_SSjz · 2024-10-30

**Soundness:** 3
**Presentation:** 4
**Contribution:** 3
**Rating:** 6
**Confidence:** 4

**Summary:**

This paper proposes a generalizable 3D Gaussian compression framework incorporated with an effective entropy minimization module inspired by 2D image compression. The main contribution of this work is compressing 3D Gaussians within a minute, using a feed-forward pipeline. To achieve this, the authors first divide Gaussian attributes into a) geometry attributes, and b) color attributes. Then, these attributes are encoded and decoded using hyper-priors from inter-/intra-Gaussian context models, along with context design from the image compression approach. Consequently, this paper achieves a compact size and comparable rendering quality compared to the existing compression method, while requiring only several seconds.

**Strengths:**

- This paper suggests a novel framework that efficiently compresses the 3D Gaussian in a feed-forward manner, whereas existing optimization-based compression methods require additional adaptation steps.

- Experimental results demonstrate that the framework achieves high compression performance with comparable rendering quality with existing compression methods.

- The authors validate that the proposed hyper-prior design, including inter-/intra-Gaussian context models, further improves the compression performance.

**Weaknesses:**

- Despite the effectiveness, the design of context models comes from heuristic priors that the Gaussians share inter-/intra-information. It could be helpful to show those relations of Gaussians to understand the justification of the proposed hyper-prior designs.
- There is no ablation on the strategy for dividing chunks in the inter-/intra-Gaussian context model. It could be helpful to show the results for several splitting designs.
- Due to arithmetic coding-based compression, it might yield notable computational costs to encode and decode the parameters.

**Questions:**

- How much time is needed for encoding and decoding parameters?
- It could be helpful to add compression performance for each attribute by demonstrating the storage size of each attribute before and after compression. (or storage size of color attributes and geometry attributes)
- How many scenes are required to acquire a feasible compression performance?
- An optimization-based compression work, ContextGS [1], has already applied an image compression-based pipeline for 3DGS using hyper-prior similar to the design of inter-Gaussian context models. However, the authors do not need to consider it, since ContextGS has not been published yet, accepted to NeurIPS 2024. Therefore, it would be helpful to mention this paper as a concurrent work.


(+minor question)
- Do you have any plans to release DL3DV-GS, which is a training dataset for this work? This could be helpful for various generalizable 3DGS approaches, including compression.


---
### Reference
[1] Wang et al., ContextGS: Compact 3D Gaussian Splatting with Anchor Level Context Model, NeurIPS 2024

---

> ### Author Response · Authors · 2024-11-22
> **Response to Reviewer 3 (SSjz)**
>
> **W1: Show relations of Gaussians.**
>
> Thanks for your suggestion. To show relations of Gaussians, we visualize bit allocation conditions of both inter- and intra-Gaussian context models. Please refer to **Appendix Section J** for a comprehensive analysis. Below, we provide an example table of the result, which shows per-parameter bit allocation conditions of color attributes ($m=1$) over different parts using context models. This table highlights the effectiveness of the context models.
>
> - For the inter-Gaussian context model, from a statistical perspective, each Gaussian in different batches is expected to hold similar amounts of information on average due to the random splitting strategy, which ensures an even distribution of Gaussians across the 3D space. Thus, as batches progress deeper (*i.e.*, B1 $\rightarrow$ B4), the bits per parameter decrease. The most significant reduction occurs between B1 and B2, since B1 lacks inter-Gaussian context, leading to less accurate probability estimates. For subsequent batches, the reduction is less pronounced, as B1 already includes $1/6$ of the total Gaussians, providing sufficient context for accurate prediction. Adding more Gaussians from later batches yields diminishing benefits. This finding validates our approach of splitting batches with varying proportions of Gaussians, where the first two batches contain fewer Gaussians.
> - For the intra-Gaussian context model, different from that in the inter-Gaussian context model, the information distribution among different chunks is not guaranteed to be equal, as the latent space is derived from a learnable MLP. As a result, C1 receives the least information because it lacks intra-Gaussian context, leading to less accurate probability predictions. Therefore, allocating more information to C1 would make entropy reduction challenging. To counter this, the neural network compensates by assigning minimal information to C1, thereby saving bits. Conversely, C2 receives the most information as it plays a pivotal role in predicting subsequent chunks (*i.e.*, C3 and C4), while its own entropy can be reduced by leveraging C1. As a result, the bits per parameter decrease consistently from C2 to C4 due to the increasingly rich context.
>
> A full comprehensive analysis of more tables and discussions covering all attributes (color and geometry) can be found in **Appendix Section J**.
>
> **Table 1:**  Per-parameter bit consumption of color attributes ($m$=1) over different parts using context models. Values are scaled by 100x for readability. B1 $\rightarrow$ B4 and C1 $\rightarrow$ C4 indicate batches and chunks of inter- and intra-Gaussian context models, respectively.
>
> |                            | B1    | B2    | B3    | B4    |
> |----------------------------|-------|-------|-------|-------|
> | C1                         | 2.45  | 2.24  | 2.21  | 2.20  |
> | C2                         | 11.19 | 10.27 | 10.16 | 10.09 |
> | C3                         | 8.94  | 8.15  | 8.10  | 8.08  |
> | C4                         | 5.80  | 5.61  | 5.59  | 5.59  |
>
>
> **W2: Ablation over the strategy for dividing chunks.**
>
> Thank you for your suggestion. We have conducted ablation studies on the split-dividing strategies for both inter- and intra-Gaussian context models and included a detailed analysis in **Appendix Section I**. Increasing the number of splits does not consistently lead to improved performance and may increase the model’s complexity. Our experiments confirm that the default splitting strategy strikes a good balance between performance and complexity. For a more comprehensive discussion, please refer to **Appendix Section I**. Below are the tables of the ablation studies for both inter- and intra-Gaussian context models.
>
> ---
>
> **Table 2:** Ablation study on the number of splits in the **inter-Gaussian context model** on the *DL3DV-GS* dataset with $\lambda=1e-4$.
>
> | Split Batches             | SIZE (MB) ↓ | PSNR (dB) ↑ | SSIM ↑  | LPIPS ↓ |
> |---------------------------|--------------|--------------|----------|----------|
> | 95%, 5% (2)              | 28.685       | 29.264       | 0.897    | 0.148    |
> | 25%, 25%, 50% (3)        | 27.667       | 29.264       | 0.897    | 0.148    |
> | 17%, 17%, 33%, 33% (4)   | 27.440       | 29.264       | 0.897    | 0.148    |
> | 13%, 13%, 25%, 25%, 25% (5) | 27.379    | 29.264       | 0.897    | 0.148    |
>
> ---
>
> **Table 3:** Ablation study on the number of splits in the **intra-Gaussian context model** on the *DL3DV-GS* dataset with $\lambda=1e-4$.
>
> | Split Chunks | SIZE (MB) ↓ | PSNR (dB) ↑ | SSIM ↑  | LPIPS ↓ |
> |--------------|--------------|--------------|----------|----------|
> | 2            | 28.562       | 29.267       | 0.897    | 0.148    |
> | 4            | 27.440       | 29.264       | 0.897    | 0.148    |
> | 8            | 27.516       | 29.264       | 0.897    | 0.148    |
> | 16           | 27.649       | 29.265       | 0.897    | 0.148    |

---

> ### Author Response · Authors · 2024-11-22
> **Response to Reviewer 3 (SSjz)**
>
> **W3&Q1: Regarding the coding efficiency of arithmetic coding.**
>
> Thanks for your suggestion. We have implemented a CUDA-based arithmetic codec which is fast. The total time for encoding is presented in Figure 4. In **Appendix Section H**, we now provide a comprehensive analysis of both the encoding and decoding time of each single component. Here are the two tables in **Appendix Section H**, which indicate **Encoding time** and **Decoding time**, respectively. During encoding, the GPCC takes most of the time to compress Gaussian locations due to its complex RD search process. During decoding, the model is much faster. Please refer to **Appendix Section H** for more details.
>
> **Table 4:** **Encoding time** of different components on the *DL3DV-GS* dataset,which has over $1.57$ million Gaussians on average in the testing set. All times are measured in seconds, with each component’s percentage share of the total encoding time provided in parentheses. The mask rate indicates the proportion of color attributes $f^{\text{col}}$ assigned to the $m=1$ path (*i.e.*, compressed via the autoencoder). Results are obtained using a single GPU.
>
> | $\lambda$   | Total time   | $\mu^\text{g}$      | $f^{\text{col}}$ ($m=1$) | $f^{\text{col}}$ ($m=0$) | $f^{\text{geo}}$ | Mask       | Others     | Mask rate |
> |-------------|--------------|---------------------|--------------------------|--------------------------|------------------|------------|------------|-----------|
> | $1e-4$      | 20.47        | 9.48 (46%)         | 2.81 (14%)               | 4.41 (22%)               | 2.35 (11%)       | 0.02 (0%)  | 1.41 (7%)  | 0.62      |
> | $2e-4$      | 18.33        | 9.46 (52%)         | 2.97 (16%)               | 2.63 (14%)               | 1.83 (10%)       | 0.02 (0%)  | 1.42 (8%)  | 0.67      |
> | $4e-4$      | 17.17        | 9.50 (55%)         | 3.09 (18%)               | 1.67 (10%)               | 1.46 (8%)        | 0.02 (0%)  | 1.44 (8%)  | 0.74      |
> | $8e-4$      | 16.44        | 9.49 (58%)         | 3.34 (20%)               | 1.01 (6%)                | 1.10 (7%)        | 0.02 (0%)  | 1.48 (9%)  | 0.82      |
> | $16e-4$     | 15.86        | 9.44 (60%)         | 3.21 (20%)               | 0.68 (4%)                | 1.00 (6%)        | 0.02 (0%)  | 1.51 (10%) | 0.89      |
>
> ---
>
> **Table 5:** **Decoding time** of different components on the *DL3DV-GS* dataset, which has over $1.57$ million Gaussians on average in the testing set. All times are measured in seconds, with each component’s percentage share of the total decoding time provided in parentheses. The mask rate indicates the proportion of color attributes $f^{\text{col}}$ assigned to the $m=1$ path (*i.e.*, compressed via the autoencoder). Results are obtained using a single GPU.
>
> | $\lambda$   | Total time   | $\mu^\text{g}$      | $f^{\text{col}}$ ($m=1$) | $f^{\text{col}}$ ($m=0$) | $f^{\text{geo}}$ | Mask       | Others     | Mask rate |
> |-------------|--------------|---------------------|--------------------------|--------------------------|------------------|------------|------------|-----------|
> | $1e-4$      | 15.93        | 3.53 (22%)         | 3.85 (24%)               | 5.07 (32%)               | 2.51 (16%)       | 0.02 (0%)  | 0.95 (6%)  | 0.62      |
> | $2e-4$      | 14.16        | 3.52 (25%)         | 4.21 (30%)               | 3.28 (23%)               | 2.14 (15%)       | 0.02 (0%)  | 0.99 (7%)  | 0.67      |
> | $4e-4$      | 12.84        | 3.52 (27%)         | 4.33 (34%)               | 2.16 (17%)               | 1.77 (14%)       | 0.02 (0%)  | 1.04 (8%)  | 0.74      |
> | $8e-4$      | 12.13        | 3.53 (29%)         | 4.68 (39%)               | 1.40 (12%)               | 1.39 (11%)       | 0.02 (0%)  | 1.10 (9%)  | 0.82      |
> | $16e-4$     | 11.47        | 3.52 (31%)         | 4.67 (41%)               | 0.87 (8%)                | 1.24 (11%)       | 0.02 (0%)  | 1.15 (10%) | 0.89      |

---

> ### Author Response · Authors · 2024-11-22
> **Response to Reviewer 3 (SSjz)**
>
> **Q2: storage size of each attribute.**
>
> Thanks for your suggestion. We now provide a detailed analysis of the storage size for each attribute in **Appendix Section G**. This includes two tables: one showing the total storage size of each attribute and the other detailing the per-parameter bits, offering a comprehensive breakdown. These two tables also allow for the calculation of the compression ratio for each component before and after compression. With increasing $\lambda$, the sizes of both color and geometry attributes decrease. Within the color attributes, the conditions for paths $m=1$ and $m=0$ differ due to the rising mask rate as $\lambda$ increases. A more comprehensive analysis is available in **Appendix Section G**.
>
> **Table 6:** **Storage size** of different components on the *DL3DV-GS* dataset. All sizes are measured in **MB**. The mask rate indicates the proportion of color attributes $f^{\text{col}}$ assigned to the $m=1$ path (*i.e.*, compressed via the autoencoder).
>
> | $\lambda$   | Total size   | $\mu^\text{g}$ | $f^{\text{col}}$ ($m=1$) | $f^{\text{col}}$ ($m=0$) | $f^{\text{geo}}$ | Mask size  | Mask rate |
> |-------------|--------------|----------------|--------------------------|--------------------------|------------------|------------|-----------|
> | $1e-4$      | 27.44        | 3.25           | 2.61                     | 10.92                    | 10.50            | 0.17       | 0.62      |
> | $2e-4$      | 24.15        | 3.25           | 2.72                     | 8.10                     | 9.93             | 0.16       | 0.67      |
> | $4e-4$      | 21.12        | 3.25           | 2.77                     | 5.67                     | 9.29             | 0.14       | 0.74      |
> | $8e-4$      | 18.14        | 3.25           | 3.01                     | 3.42                     | 8.34             | 0.12       | 0.82      |
> | $16e-4$     | 15.83        | 3.25           | 2.94                     | 1.76                     | 7.80             | 0.08       | 0.89      |
>
> ---
>
> **Table 7:** **Per-parameter bits** of different components on the *DL3DV-GS* dataset. All sizes are measured in **bit**. The mask rate indicates the proportion of color attributes $f^{\text{col}}$ assigned to the $m=1$ path (*i.e.*, compressed via the autoencoder).
>
>
> | $\lambda$   | Weighted AVG | $\mu^\text{g}$ | $f^{\text{col}}$ ($m=1$) | $f^{\text{col}}$ ($m=0$) | $f^{\text{geo}}$ | Mask size  | Mask rate |
> |-------------|--------------|----------------|--------------------------|--------------------------|------------------|------------|-----------|
> | $1e-4$      | 2.51         | 5.92           | 0.45                     | 3.29                     | 6.94             | 0.94       | 0.62      |
> | $2e-4$      | 2.21         | 5.92           | 0.43                     | 2.92                     | 6.56             | 0.89       | 0.67      |
> | $4e-4$      | 1.94         | 5.92           | 0.40                     | 2.58                     | 6.14             | 0.81       | 0.74      |
> | $8e-4$      | 1.67         | 5.92           | 0.39                     | 2.32                     | 5.51             | 0.66       | 0.82      |
> | $16e-4$     | 1.45         | 5.92           | 0.36                     | 2.10                     | 5.15             | 0.48       | 0.89      |

---

> ### Author Response · Authors · 2024-11-22
> **Response to Reviewer 3 (SSjz)**
>
> **Q3: Number of scenes to acquire a feasible compression performance.**
>
> We design FCGS to train on massive 3DGS representations to acquire sufficient prior information for compression, enabling it to conduct a feedforward-based compression using the learned prior information. Although preparing this training data is time-intensive, it is a one-time investment—once trained, FCGS can be directly applied for compression without requiring further optimization, making the effort highly valuable. To evaluate FCGS’s performance when trained on significantly less data, we conducted experiments using only $100$ scenes, as shown in the table below. Even with this reduced training set, FCGS still delivers strong results, achieving comparable fidelity metrics with only $12.9$% increase in size. This robustness is due to FCGS’s compact and efficient architecture, with a total model size of less than $10$ MB. Discussions of this evaluation has been added to **Appendix Section L**.
>
> ---
> **Table 8:** Results of **training with less data**. Experiments are conducted on the *DL3DV-GS* dataset with $\lambda=1e-4$.
>
> | \# Training data           | SIZE (MB) ↓ | PSNR (dB) ↑ | SSIM ↑ | LPIPS ↓ |
> |----------------------------|-------------|-------------|--------|---------|
> | Full (\# $6670$ scenes)       | 27.440      | 29.264      | 0.897  | 0.148   |
> | Subset (\# $100$ scenes)    |  30.981  |   29.256   |  0.897  | 0.148  |
>
> **Q4: Regarding ContextGS.**
>
> Thanks for your advice.
> ContextGS proposes structuring anchors into multiple levels to model entropy. However, its anchor location quantization process introduces deviations, which impacts fidelity and requires finetuning. Differently, our inter-Gaussian context model overcomes this limitation by creating grids in a feed-forward manner for context modeling, without modifying Gaussian locations, eliminating the need of finetuning.
> We have now added this discussion to **Line 262-266**.
>
> **Q5: Open-source of the DL3DV-GS dataset.**
>
> The *DL3DV-GS* dataset will be open-source. However, the *DL3DV-GS* dataset is derived from *DL3DV*, which requires access permissions. To facilitate open-source access to *DL3DV-GS*, we will coordinate with the *DL3DV* authors to establish a legal and compliant method for release.
>
> **If you have any further questions or concerns, please feel free to reach out to us for discussion.**

---

> > ### Comment · Reviewer_SSjz · 2024-11-26
> >
> > I appreciate the authors for addressing all my concerns during the discussion period. I will maintain my positive rating.

---

> > > ### Author Response · Authors · 2024-11-26
> > > **Response to Reviewer 3 (SSjz)**
> > >
> > > Thank you for your thoughtful engagement and valuable suggestions, which have been instrumental in improving our paper. We sincerely appreciate your support and the time you dedicated to reviewing our work!

---

### Official Review · Reviewer_VFpS · 2024-11-01

**Soundness:** 3
**Presentation:** 3
**Contribution:** 2
**Rating:** 6
**Confidence:** 3

**Summary:**

This paper presents a novel, optimization-free compression pipeline for 3D Gaussian Splatting (3DGS) representations, called FCGS (Fast Compression of 3D Gaussian Splatting). FCGS uses a single feed-forward model to compress 3DGS representations, drastically reducing compression time compared to pruning and optimization-based approaches. The paper further details the Multi-path Entropy Module (MEM) and context models that FCGS employs to balance size and fidelity during compression. Finally, the authors demonstrate the efficacy of FCGS through extensive experiments, achieving a compression ratio exceeding 20x while preserving high fidelity.

**Strengths:**

- The paper provides a complete overview of related works on 3DGS with compression and structure it according to the methodology, see Section 1 and 2.
- The overall goal of the paper is clear and the authors motivate compression potential for 3DGS scenes well.
- The quantitative evaluation is very good, as the authors provide a comparison to several relevant methods, like light gaussians and Niedermayr etal.,  on three different datasets (MipNeRF360, T&T and DL3DV-GS) and present results in expressive graphs that highlight the trade-off of size and quality, see Figure 4.
- In addition to comparisons to compression works, the author further extent t0 feed forward reconstruction models and find a more efficient representation of their approach.
- Overall the experiments indicate that the proposed methods is an effective approach for compression performs favourable to most other approaches in terms of (PSNR/SIZE), however less effective than, Niedermayr et al.. See Figure 4.

**Weaknesses:**

- The methodology is very complex and the respective section in the paper is hard to understand even for someone familiar with 3DGS. The reason is that there is information missing at the appropriate position in the text. For example, figure 2 in line 262 is is unclear how the Gaussians are divided into batches (randomly, per grid cell stays unclear). Hence it appears to hard to reproduce. Please see the Questions sections for more open points.
- The method's focus lies very much on increasing the compression speed and it does not improve over existing methods in terms of PSNR/Size, see Figure 4 (Simon*). One could argue that there is limited interests in compression speed, since a 3D GS reconstruction takes several minutes to hours and a follow up compression of a few minutes (e.g. Niedermayr et al.) does not impact the overall reconstruction time significantly.
- For qualitative evaluation, I'd have expected more visual comparisons to baseline methods. There is only comparisons to 3DGS in paper and supplementary material. Please provide more qualitative evidence.

**Questions:**

- What are the analysis transform and synthesis transform exactly? Architecture? Space?
- Why can "the potential nonlinearity of the MLPs still cause considerable deviations in decoded gaussians", line 222?
- How are the the set of Gaussian split and what is the criterion to split them?

**Details Of Ethics Concerns:**

No concerns

---

> ### Author Response · Authors · 2024-11-22
> **Response to Reviewer 2 (VFpS)**
>
> **W1: Complexity of the method.**
>
> Our method is well structured for compression, which consists of both MEM and context models to achieve efficient compression. MEM selectively projects Gaussians to a latent space to balance the fidelity and size; While we adapt the common idea of context models in traditional data compression to 3DGS compression, which uses decoded parameters as context to predict remaining parameters. Both MEM and context models are critical and serve for different purposes to achieve a good compression performance, as stated by Reviewer 1 (xSne): “It introduces a Multi-path Entropy Module (MEM) to adaptive control the compression size and quality. It also designs inter- and intra-Gaussian context models to enhance compression efficiency.” Moreover, the batch dividing strategy has been stated in **Line 370** in the implementation details, and additional ablation studies on various splitting strategies for our context models are presented in **Appendix Section C** and **Appendix Section I**. Overall, rather than adding complexity, these two components are methodically organized to achieve effective compression performance.
>
> **W2: Compression speed versus PSNR/size.**
>
> As discussed in **Line 82-88**, the two pipelines of per-scene optimization-based compression (ours) and generalizable optimization-free compression (others) cater to distinct use cases. Selecting the appropriate compression pipeline depends on specific application requirements. For scenarios where compression speed is less critical (*i.e.*, your example), such as offline processing, optimization-based compression can be advantageous. However, FCGS supports the compression of 3DGS from both optimization and feed-forward models, as demonstrated in Subsection 4.2. **Moreover, generalizable optimization-free compression offers many other advantages**:
> - Independence from ground-truth supervision: Unlike optimization-based approaches that rely on costly, image-based guidance, optimization-free methods operate directly on 3DGS representations without the need for such  supervision.
> - Speed sensitivity: This is crucial for 3DGS generated by feed-forward models. Such models produce 3DGS rapidly, and applying per-scene optimization compression in these 3DGSes incurs significant computational and time overhead.
> - Suitability for lightweight devices: Devices like mobile phones lack the computational power for resource-intensive per-scene optimization.
>
> Overall, our FCGS can operate in all situations and its advantages become more pronounced when compressing 3DGS generated by feed-forward models.
>
> **W3: More qualitative evaluation compression to other baselines.**
>
> Thank you for your suggestion. We have incorporated additional baseline methods for qualitative comparison:
> - Figure 5 in the main paper now includes qualitative results of the baselines.
> - Figure E in **Appendix Section N** further extends these comparisons.
>
> These updates can be found in the revised version of the paper. Additionally, we have added a note in the caption of Figure 5 to guide readers to Figure E in **Appendix Section N** for a more detailed qualitative comparison.
>
> **Q1: Analysis transform and synthesis transform.**
>
> They are implemented as simple MLPs: $g_a$ and $g_s$ consist of 4 layers each, while $h_a$ and $h_s$ have 3 layers each. This description has now been added to the caption of Figure 2.
>
> **Q2: Potential nonlinearity of MLPs.**
>
> The forward pass of MLPs is inherently non-invertible due to non-linear activation functions, which means that projecting an input (*i.e.*, $x$) into latent space (*i.e.*, $y$) via an MLP does not allow for exact recovery of the original input from the latent. As a result, MLP-decoded attributes (*i.e.*, $\hat{x}$) cannot always align precisely with the original attributes (*i.e.*, $x$) for each Gaussian. This misalignment would be amplified in rendering, leading to a significant fidelity drop in the rendered images. We have clarified and polished the writing in **Line 221-226**.
>
> **Q3: Split of Gaussians.**
>
> The split criterion is in the implementation details (**Line 370**), where we utilize a random split with 4 steps. Note that a consistent randomness is applied between the encoder and decoder to ensure accurate decoding. Importantly, this randomness minimally impacts performance—experiments with different seeds yielded similar results, as discussed in **Appendix Section C**. We also tested Farthest Point Sampling (FPS) for splitting but found that it underperformed our random splitting, likely because FPS does not distribute samples evenly. Furthermore, as discussed in **Appendix Section I**, we explored various numbers of splits; While more steps bring slight benefits, 4 steps already achieve strong performance.
>
> **If you have any further questions or concerns, please feel free to reach out to us for discussion.**

---

> > ### Comment · Reviewer_VFpS · 2024-11-25
> >
> > Thank you for the detailed comments. Your responses resolved most of my concerns (W1,W3,Q1,Q2,Q3). Regarding W2, I'm still questioning the usefulness of generalizable optimization-free compression if it is worse than the optimization based approaches. However, forward-looking, I see potential impact in combination with feed-forward 3DGS methods. Hence, I increase my rating to 6.

---

> > > ### Author Response · Authors · 2024-11-25
> > > **Response to Reviewer 2 (VFpS)**
> > >
> > > Thank you for your thoughtful engagement and valuable suggestions, which have been instrumental in improving our paper. We sincerely appreciate your support and the time you dedicated to reviewing our work!

---

### Official Review · Reviewer_xSne · 2024-11-01

**Soundness:** 3
**Presentation:** 3
**Contribution:** 3
**Rating:** 8
**Confidence:** 4

**Summary:**

The paper proposes a new compression model for 3D Gaussian Splatting (3DGS) representations. It aims to reduce compression time in an optimization-free pipeline, which is different from existing work. It introduces a Multi-path Entropy Module (MEM) to adaptive control the compression size and quality. It also designs inter- and intra-Gaussian context models to enhance compression efficiency. Experiments on multiple datasets show a large compression rate and high image quality.

**Strengths:**

1. It proposes a novel optimization-free pipeline for 3DGS-based compression.
2. MEM module to adaptively control size and fidelity.
3. It also designs inter- and intra-Gaussian context models to enhance compression efficiency.
4. The proposed method achieves a high compression rate and surpasses the existing optimized-based methods in various experiments.

**Weaknesses:**

1. The training data is very large, and it takes 60 GPU days to get. Per-scene optimization does not need such an amount of training data. Will smaller data hurt the effectiveness of the feedforward model?
2. More discussion of failure cases. it could be useful to discuss / show scenarios where the method performs poorly.

**Questions:**

1. Can authors provide visualization of inter- and intra-Gaussian context models on real input data. It would help reader better understand how this method work.

2. Will the code or data open-source for reproduction ? the training set is interesting and would benefit many other tasks.

---

> ### Author Response · Authors · 2024-11-22
> **Response to Reviewer 1 (xSne)**
>
> **W1: Amount of the Training data.**
>
> We design FCGS to train on massive 3DGS representations to acquire sufficient prior information for compression, enabling it to conduct a feedforward-based compression using the learned prior information. Although preparing this training data is time-intensive, it is a one-time investment—once trained, FCGS can be directly applied for compression without requiring further optimization, making the effort highly valuable. To evaluate FCGS’s performance when trained on significantly less data, we conducted experiments using only $100$ scenes, as shown in the table below. Even with this reduced training set, FCGS still delivers strong results, achieving comparable fidelity metrics with only $12.9$% increase in size. This robustness is due to FCGS’s compact and efficient architecture, with a total model size of less than $10$ MB. Discussions of this evaluation has been added to **Appendix Section L**.
>
> ---
> **Table 1:** Results of **training with less data**. Experiments are conducted on the *DL3DV-GS* dataset with $\lambda=1e-4$.
>
> | \# Training data           | SIZE (MB) ↓ | PSNR (dB) ↑ | SSIM ↑ | LPIPS ↓ |
> |----------------------------|-------------|-------------|--------|---------|
> | Full (\# $6670$ scenes)       | 27.440      | 29.264      | 0.897  | 0.148   |
> | Subset (\# $100$ scenes)    |  30.981  |   29.256   |  0.897  | 0.148  |
>
> **W2: Analysis of failure cases.**
>
> Thanks for pointing this out. Failure cases and limitations of FCGS had been discussed in **Appendix Section B**. We now add indicators in the main paper to this content and also polish it for clearer expression. Failures mainly arise from incorrect selections of $m$ by MEM due to the domain gap: a Gaussian that should have been assigned to the $m=0$ path for fidelity preservation may be mistakenly assigned to $m=1$, and the autoencoder in this path fails to recover these Gaussian values properly from their latent space, leading to significant fidelity degradation. This issue becomes noticeable for 3DGS from feed-forward models such as MVSplat [1], due to a distribution gap between our training data (3DGS from optimization) and those from feed-forward models like MVSplat [1]. We mitigate this problem by enforcing all Gaussians to the $m=0$ path for compressing 3DGS from feed-forward models, thereby prioritizing fidelity. For instance, to compress 3DGS from MVSplat, $m$ deduced from MEM may not be correct, which would result in a PSNR drop of approximately $0.7$dB at a slightly smaller compression size compared to setting all $m$ as 0s.
> Overall, this is our major limitation, which points out an interesting problem for future work: __*how to design optimization-free compression models that can be well generalized to 3DGS with different value characteristics*__. Please refer to **Appendix Section B** for a more detailed analysis.

---

> ### Author Response · Authors · 2024-11-22
> **Response to Reviewer 1 (xSne)**
>
> **Q1: Visualization of inter- and intra-Gaussian context models.**
>
> For visualization of inter- and intra-Gaussian context models, please refer to **Appendix Section J** for a comprehensive analysis. Below, we provide an example table of the result, which shows per-parameter bit allocation conditions of color attributes ($m=1$) over different parts using context models. This table highlights the effectiveness of the context models.
>
> - For the inter-Gaussian context model, from a statistical perspective, each Gaussian in different batches is expected to hold similar amounts of information on average due to the random splitting strategy, which ensures an even distribution of Gaussians across the 3D space. Thus, as batches progress deeper (*i.e.*, B1 $\rightarrow$ B4), the bits per parameter decrease. The most significant reduction occurs between B1 and B2, since B1 lacks inter-Gaussian context, leading to less accurate probability estimates. For subsequent batches, the reduction is less pronounced, as B1 already includes $1/6$ of the total Gaussians, providing sufficient context for accurate prediction. Adding more Gaussians from later batches yields diminishing benefits. This finding validates our approach of splitting batches with varying proportions of Gaussians, where the first two batches contain fewer Gaussians.
> - For the intra-Gaussian context model, different from that in the inter-Gaussian context model, the information distribution among different chunks is not guaranteed to be equal, as the latent space is derived from a learnable MLP. As a result, C1 receives the least information because it lacks intra-Gaussian context, leading to less accurate probability predictions. Therefore, allocating more information to C1 would make entropy reduction challenging. To counter this, the neural network compensates by assigning minimal information to C1, thereby saving bits. Conversely, C2 receives the most information as it plays a pivotal role in predicting subsequent chunks (*i.e.*, C3 and C4), while its own entropy can be reduced by leveraging C1. As a result, the bits per parameter decrease consistently from C2 to C4 due to the increasingly rich context.
>
> A full comprehensive analysis of more tables and discussions covering all attributes (color and geometry) can be found in **Appendix Section J**.
>
> **Table 2:**  Per-parameter bit consumption of color attributes ($m$=1) over different parts using context models. Values are scaled by 100x for readability. B1 $\rightarrow$ B4 and C1 $\rightarrow$ C4 indicate batches and chunks of inter- and intra-Gaussian context models, respectively.
>
> |                            | B1    | B2    | B3    | B4    |
> |----------------------------|-------|-------|-------|-------|
> | C1                         | 2.45  | 2.24  | 2.21  | 2.20  |
> | C2                         | 11.19 | 10.27 | 10.16 | 10.09 |
> | C3                         | 8.94  | 8.15  | 8.10  | 8.08  |
> | C4                         | 5.80  | 5.61  | 5.59  | 5.59  |
>
> **Q2: Open-source of the code and dataset.**
>
> Both the code and the *DL3DV-GS* dataset will be open-source. The *DL3DV-GS* dataset is derived from *DL3DV*, which requires access permissions. To facilitate open-source access to *DL3DV-GS*, we will coordinate with the *DL3DV* authors to establish a legal and compliant method for release.
>
> **If you have any further questions or concerns, please feel free to reach out to us for discussion.**
>
> **Reference:**
>
> [1] Chen Y, Xu H, Zheng C, et al. Mvsplat: Efficient 3d gaussian splatting from sparse multi-view images[C] ECCV 2024.

---

> > ### Comment · Reviewer_xSne · 2024-11-26
> >
> > The rebuttal solves my concerns, i will raise my score.

---

> > > ### Author Response · Authors · 2024-11-26
> > > **Response to Reviewer 1 (xSne)**
> > >
> > > Thank you for your thoughtful engagement and valuable suggestions, which have been instrumental in improving our paper. We sincerely appreciate your support and the time you dedicated to reviewing our work!

---

### Author Response · Authors · 2024-11-22
**General Response**

We sincerely thank all reviewers for their detailed and constructive comments. We are encouraged to see that, in general, clear motivations and novel technical contributions are recognized by reviewers: “It proposes a novel optimization-free pipeline for 3DGS-based compression.” (Reviewer xSne), “This paper presents a novel, optimization-free compression pipeline … The overall goal of the paper is clear and the authors motivate compression potential for 3DGS scenes well.” (Reviewer VFpS), “This paper suggests a novel framework that efficiently compresses the 3D Gaussian in a feed-forward manner” (Reviewer SSjz), “This article provides a novel approach to context modeling for exploring the correlations between Gaussians.” (Reviewer SkMb).

Based on the suggestions provided, we have now offered comprehensive and thorough responses. Below, we summarize the content of our responses:

- **Clarity Issues**: For concerns regarding unclear expressions in equations, figures, sentences, or paragraphs, we have provided one-to-one responses addressing each issue raised.
- **Missing Experiments or Visualizations**: Where questions about additional experiments or visualization results were raised, we have addressed them with concise explanations in the following chatboxes. Furthermore, we have included comprehensive analyses in newly **added sections of the Appendix** in the revised paper, and you can refer to it for more detailed results.
- **Other Questions**: For all other concerns, we have provided detailed responses one by one.

For revisions made to the paper, we have updated the content in the revised version and **marked changes in blue** for visibility (including updates to the main paper and the Appendix in the supplementary materials). __*Unless otherwise specified, all Line, Figure and Section numbers mentioned in the following responses refer to the revised version of the paper and Appendix.*__ We hope these answers and the revisions have properly addressed all the review concerns and will be grateful to see further feedback from the reviewers.

---

### Author Response · Authors · 2024-11-24
**Kind Request for Discussions and Feedback for Paper 702**

Dear Reviewers,

We would like to express our sincere gratitude for the time and effort you have dedicated to reviewing our paper and for sharing your valuable suggestions.

As the discussion phase approaches its conclusion, we hope that our responses have sufficiently addressed your concerns and questions.

Your thoughtful feedback plays a crucial role in refining our work, and we deeply appreciate your insights. We look forward to your continued engagement and any further thoughts you may wish to share.

Thank you once again for your time and consideration.

Best regards,

The Authors of Paper 702

---

### Meta-Review · Area_Chair_sk1U · 2024-12-20

**Metareview:**

This paper presents an efficient 3D Gaussian Splatting (3DGS) compression method. The proposed approach is novel, offering an optimization-free pipeline for 3DGS compression in a feed-forward manner. A multi-path entropy module controls the tradeoff between size and quality, while inter- and intra-Gaussian context models further enhance compression performance. The method demonstrates strong compression rates across multiple datasets while significantly reducing execution time.

The main weakness is that the compression size and quality may not surpass optimization-based methods. However, the proposed method compresses 3DGS in a feed-forward manner and reduces per-scene compression time. It is the first generalizable optimization-free 3DGS compression method and offers potential applications that previous methods cannot address.

**Additional Comments On Reviewer Discussion:**

Several issues were raised in the initial reviews, and the rebuttal effectively addressed most of them.

First, the primary goal of the proposed method is to accelerate compression. While there were concerns about whether compression time is a critical factor compared to size and quality, the rebuttal clarified that some applications prioritize speed over size or quality. It also emphasized that the method’s feed-forward approach offers unique advantages. In addition, direct comparisons with optimization-based methods are somewhat unfair. This also addressed the concern about comparisons being limited to methods with worse rendering quality.

Second, a significant concern was the large amount of training data required, which could take considerable time to prepare and train. The rebuttal argued this is a one-time effort, as subsequent applications do not require expensive optimization. Furthermore, an additional experiment demonstrated that substantially reduced training data can still deliver reasonable performance. This also responded to another review's question about the number of scenes needed to acquire a feasible compression performance.

Finally, some reviewers requested visualizations and an ablation study related to the inter-/intra-Gaussian context model. The rebuttal addressed this by providing them in the appendix.

---

### Decision · Program_Chairs · 2025-01-22

Accept (Poster)